# Wind Farm Cable Connection Layout Optimization Using a Genetic Algorithm and Integer Linear Programming

Eduardo J. Solteiro Pires [1,2], Adelaide Cerveira [2,3] and José Baptista [1,2,*]

1   Department of Engineering, University of Trás-os-Montes and Alto Douro, 5000-801 Vila Real, Portugal; epires@utad.pt
2   INESC-TEC UTAD's Pole, Quinta de Prados, 5000-801 Vila Real, Portugal; cerveira@utad.pt
3   Department of Mathematics, University of Trás-os-Montes and Alto Douro, 5000-801 Vila Real, Portugal
*   Correspondence: baptista@utad.pt

**Abstract:** This work addresses the wind farm (WF) optimization layout considering several substations. It is given a set of wind turbines jointly with a set of substations, and the goal is to obtain the optimal design to minimize the infrastructure cost and the cost of electrical energy losses during the wind farm lifetime. The turbine set is partitioned into subsets to assign to each substation. The cable type and the connections to collect wind turbine-produced energy, forwarding to the corresponding substation, are selected in each subset. The technique proposed uses a genetic algorithm (GA) and an integer linear programming (ILP) model simultaneously. The GA creates a partition in the turbine set and assigns each of the obtained subsets to a substation to optimize a fitness function that corresponds to the minimum total cost of the WF layout. The fitness function evaluation requires solving an ILP model for each substation to determine the optimal cable connection layout. This methodology is applied to four onshore WFs. The obtained results show that the solution performance of the proposed approach reaches up to 0.17% of economic savings when compared to the clustering with ILP approach (an exact approach).

**Keywords:** wind farm; cable connection layout; genetic algorithms; integer linear programming

## 1. Introduction

With the expected increase in world energy demand, access to reliable energy at affordable prices is essential for economic and social well-being and is an important development indicator. At the same time, energy production lies at the root of the pollution and greenhouse gas emissions that contribute decisively to climate change. Thus, the fight against climate change and the reduction in greenhouse gas emissions became a central issue on the agendas of practically all countries in the world, with issues related to energy sources and energy efficiency fundamental to this cause. Kabouris and Kanellos [1] present significant technical challenges posed by the integration of renewable energy mainly due to its variable and hard-to-predict nature.

Currently, around 85% of the world's primary energy consumption comes from non-renewable energy sources, with renewable sources representing only 15%, and of these, wind energy represents 2.1% [2]. Therefore, there is still a long way to go, which involves continuous renewable energy investments, particularly onshore and offshore wind power. The vast majority of medium and large wind farms are constituted of several dozen wind turbines dispersed by agglomerates that can be connected to one or more substations regarding onshore wind farms. In this type of situation, the technical solutions found on the ground to interconnect all turbines and substations can be diverse, resulting in different costs of installing the distribution network and different values for energy losses.

Optimizing the wind farm distribution grid is crucial for several reasons, and it contributes to the overall efficiency, reliability, and economic viability of the wind energy system. The best optimization solutions can maximize energy production, extracting the

maximum amount of energy from wind resources. The grid's stability and reliability can be enhanced, and efficiency improvements resulting from optimization can lead to cost savings. This includes better maintenance planning, reduced downtime, and increased lifespan of equipment. Additionally, optimized energy production can contribute to a more cost-effective energy generation process. Many regions have regulations and standards in place to ensure the stability and reliability of the power grid. Optimizing the wind farm power grid helps meet these regulatory requirements, avoiding penalties and ensuring compliance. In summary, optimizing the power grid of a wind farm is essential for maximizing energy production, ensuring grid stability, reducing costs, meeting regulatory requirements, and advancing the overall sustainability and reliability of the energy system.

Most of the works found in the literature consider wind farms with only one substation and optimize the cable layout to interconnect the turbines to the substation, using exact methods [3–11] or meta-heuristics [12–17]. On the other hand, when several substations are taken into account, the topology connection (identify the substation at which the turbines are connected) and the cable connection (connection layout between turbines and its substation) designs are considered separately, leading to suboptimal solutions. The last kind of work is described in the following.

Fischetti and Pisinger [18] combine mixed-integer linear programming with math-heuristics to optimize the cable connections of wind farms. The problem was modeled to consider more than one substation. However, the MILP model only manages to solve the smallest problems in a reasonable time. Since the instances were not clearly described in the work, it is not possible to know if the results consider windfarms with more than one substation.

In [19], the authors propose an integer linear programming model for the design of wind farms with multiple substations, minimizing the costs of infrastructure and energy losses. Moreover, considering a discrete set of possible turbine locations, the model is able to identify those that should be present in the optimal solution, hence addressing the optimal location of the substation(s) in the wind farm.

Srikakulapu and U [20] minimize the investment cost and power losses in the cable connections of wind farms using a three-step algorithm: wind turbines allocation, where the turbines are grouped using a fuzzy clustering algorithm; wind turbines reallocation, where the turbines are allocated to their nearest substations using a binary programming model; cable layout optimization, where a minimum spanning tree algorithm is used to minimize the total length layout design for wind farm cable connections. Considered separately, the allocation turbines and the optimization layout length could lead to a suboptimal solution. They present results for a wind farm with three substations and 50 turbines. Zuo et al. [21] use a fuzzy clustering technique to determine the substation locations and the minimum spanning tree model to find the cable connection layout for offshore wind farms' re-powering and expansion.

Dutta and Overbye [22] use a clustering algorithm to determine the cable layout for wind farms. In this work, the authors claim that the proposed method yielded lower collector system real power losses when compared to the conventional radial or daisy chain cable layout method.

Wu and Wang [23] use the k-means clustering and ant colony algorithm to reduce wind farms' construction costs and the collector system's reliability. They present a problem with four substations, using the clustering algorithm to assign the turbines to a substation.

Wang et al. [24] present an integrated design method for wind farms, considering the substation location, connection topology, and cable cross-sections to minimize the total cost. They use an evolutionary algorithm to solve the problem. Moreover, they use a heuristic algorithm to find the substation coordinates, the substation associated with each turbine, and the cable layout. The authors consider only wind farms with one and two substations with 56 and 40 turbines, respectively.

Pillai [25] proposes an approach for a cable design of a wind farm. They divide the problem into some subproblems, each one with one substation. The substation places

are determined via the proposed approach, where a capacitated clustering approach for placing the substations is used. On the other hand, a mixed-integer linear program (MILP) to solve each subproblem and determine the cable connection and cable type to be installed is used. The MILP uses some initial solutions obtained via heuristics.

This paper's main contribution is to optimize the layout of wind farms, in one step, considering multiple substations and cable connections, in contrast to the usual approaches found in the literature, which address only one singular substation or a reduced number of turbines. In the optimization process, a genetic algorithm is used to determine the topology design, and an integer linear programming model determines the optimal cable connection. The overall objective function minimizes energy losses and cable installation costs. The case studies presented consider up to five substations and 120 wind turbines, but the methodology could be extended to higher dimensions.

Wind farms with several substations result in lower power transformers' installation and consequently, lower insulation levels in all protection equipment, which increases the efficiency of the WF, resulting in higher profitability by improving the LCOE (Levelized Cost of Electricity).

The remaining part of the paper is divided into the following. Section 2 describes the electrical power grid to model the wind farm problem. Section 3 characterizes briefly the wind farms considered in the study and the adopted methodology to solve the layout problem. Section 4 describes how the turbines' assignment to the substations was addressed and the cable connections were determined. Section 5 presents the obtained results and discusses them. Finally, Section 6 draws the main conclusions.

## 2. Electrical Power Grid Modeling

The selection of the structure of a wind farm is a process that involves multidisciplinary teams (engineering, economy, environment, and even evaluation of social impacts), with different objectives but always in search of a common goal that goes through the optimization of the whole structure harmoniously.

A wind farm consists of three blocks, the energy collection system, the integration system, and the transmission system. The first one is responsible for collecting the wind turbines' energy and routing it to the main substation. In the case of large wind farms, there may be an arrangement in clusters, each with a substation interconnected through a distribution network to the main substation. The second block is responsible for integrating all the elements, optimally allowing the energy produced to be routed to the substations, which is usually performed with voltage levels of 20/30 kV. The transmission system is responsible for injecting the produced energy into the transmission network, which is generally conducted at high voltage, for example, 150 kV.

In the wind farm distribution network, where each node represents a wind turbine, to carry out a stationary analysis of the power flow, it is essential to know all the network parameters to develop a model for all of them. These networks have a radial structure, and it is necessary to understand how to calculate the transit of energy flowing there. Some work in the literature deals precisely with the problem of power transit in radial networks [26–29]. In addition, works such as [30] are dedicated to the reactive energy optimization problem in radial networks, presenting contributions in calculating load flow and adapting to these networks' radial characters.

In this paper's study, which has as the main objective the optimization of the cable layout considering several substations in wind farms, as shown in Figure 1, it is essential to know the parameters associated with the internal distribution network.

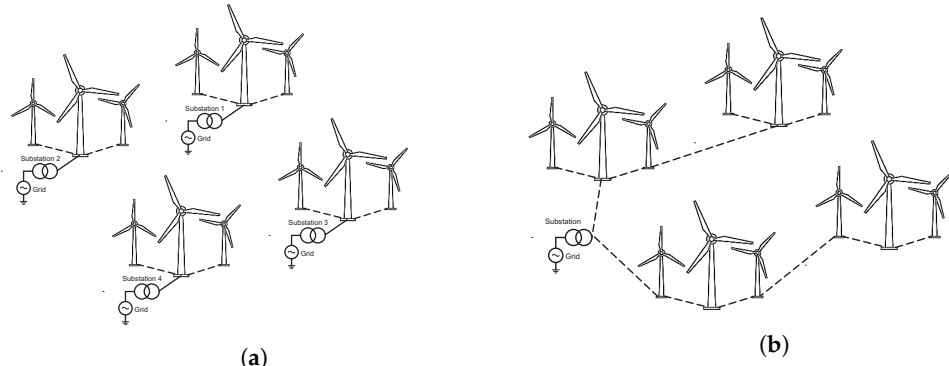

**Figure 1.** Wind farm layouts: (**a**) with several clusters and substations, (**b**) with only one substation.

As referenced by Cerveira et al. [4], in wind farms' distribution networks, the short line model should be used. This choice is because the networks that connect the various elements of a wind farm are lower than a few kilometers, where the R/X ratio is high. With the short line model, several simplifications can be made, and the cables' shunt admittance can be neglected. Therefore, the network branches can be represented by the model of Figure 2. Moreover, a radial cable network structure is considered where Kirchoff's current and voltage laws and the branch current stability constraint are guaranteed.

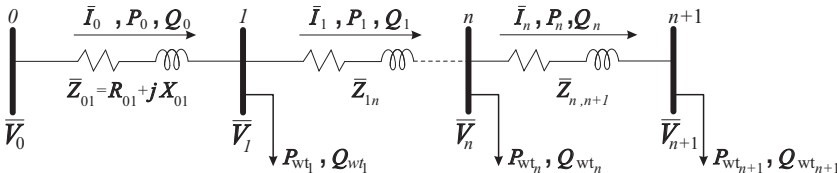

**Figure 2.** Line diagram of a radial distribution system.

The load constraints considered in this study are power balance constraints described by a set of power flow equations for balanced radial distribution networks developed in [31]. Therefore, the power flowing at the receiving end of the branch $n + 1$, $P_{n+1}$, $Q_{n+1}$, and the voltage magnitude at the sending end $V_{n+1}$ can be expressed via Equations (1)–(3).

$$P_{n+1} = P_n - P_{\mathrm{wt}_n} - R_{n,n+1} \cdot \frac{P_n^2 + Q_n^2}{|V_n|^2}, \tag{1}$$

$$Q_{n+1} = Q_n - Q_{\mathrm{wt}_n} - X_{n,n+1} \cdot \frac{P_n^2 + Q_n^2}{|V_n|^2}, \tag{2}$$

$$|V_{n+1}|^2 = |V_n|^2 - 2(R_{n,n+1}P_n + X_{n,n+1}Q_n) + (R_{n,n+1}^2 + X_{n,n+1}^2)\frac{P_n^2 + Q_n^2}{|V_n|^2}. \tag{3}$$

where $P_n$ and $Q_n$ represent the active and reactive power, respectively, leaving the sending bus $n$, while $P_{\mathrm{wt}_n}$ and $Q_{\mathrm{wt}_n}$ are the active and reactive power that flow on the bus $n$, regarding the wind farm branches connected to that bus, and $V_n$ is the voltage at bus $n$. The branch resistance and reactance between buses $n$ and $n + 1$ are represented by $R_{n,n+1}$ and $X_{n,n+1}$, respectively.

It should be noted that the voltage drop between any two buses should be within regulatory limits, which may be up to 5% of the network voltage, $U$. The model considered here does not include those constraints that are naturally guaranteed in the solutions by the dimensions of the case studies discussed.

To obtain the branch losses between the bus $n$ and bus $n + 1$, Equations (4)–(6) are used:

$$I_n^2 = \frac{P_n^2 + Q_n^2}{|V_n|^2}, \tag{4}$$

$$P_{\mathrm{loss}(n,n+1)} = R_{n,n+1} \cdot I_n^2, \tag{5}$$

$$Q_{\text{loss}(n,n+1)} = X_{n,n+1} \cdot I_n^2. \tag{6}$$

Therefore, the total power losses, $T_{\text{loss}}$, can then be obtained by adding the losses from all cable connections, as given by

$$T_{\text{loss}} = \sum_{k \in N} P_{\text{loss}(k,k+1)} + Q_{\text{loss}(k,k+1)} \tag{7}$$

where $N$ is the set of the nodes in the network.

The values of $R_{n,n+1}$ and $X_{n,n+1}$ depend on the cable used. In this work, a set of cable types, presented in Table 1, is considered. Each cable type $k$ is characterized by its section, an inductance $L_k$ per unit of length, a resistance $R_k$ per unit of length, a maximum current intensity $I_{z_k}$ that it can support, and a cost $C_k$ per unit of length.

**Table 1.** Characteristics of unipolar cables (LXHIOV) 18/30 kV.

| Type $k$ | Section (mm$^2$) | Inductance $L$ (mH/km) | Electrical Resistance $R$ (Ω/km) | Max. Current $I_z$ (A) | Price (EUR/m) |
|---|---|---|---|---|---|
| 1 | 50 | 0.62 | 0.6410 | 169 | 6.80 |
| 2 | 70 | 0.59 | 0.4430 | 207 | 7.12 |
| 3 | 95 | 0.57 | 0.3200 | 247 | 7.98 |
| 4 | 120 | 0.55 | 0.2530 | 281 | 8.70 |
| 5 | 150 | 0.54 | 0.2060 | 313 | 12.77 |
| 6 | 185 | 0.53 | 0.1640 | 354 | 13.23 |
| 7 | 240 | 0.50 | 0.1250 | 408 | 14.89 |
| 8 | 300 | 0.49 | 0.1000 | 458 | 17.50 |
| 9 | 400 | 0.47 | 0.0778 | 519 | 21.09 |
| 10 | 500 | 0.46 | 0.0605 | 585 | 23.77 |

The maximum current intensity, $I_z$, bounds the number of wind turbines in any branch line, i.e., in a set of connections starting with a direct link to the substation. Furthermore, it will determine the types of cable that could be used depending on the number of downstream wind turbines. It should be noted that the rated current drawn by each turbine, defined by $I_r$, is given by

$$I_r = \frac{P_r}{\sqrt{3} \cdot U \cdot \cos \varphi} \tag{8}$$

where $P_r$ is the rated power of the wind turbines and $U$ is the interconnection grid's voltage. The value of $\cos \varphi$ is the turbine's power factor, and, in the case studies, it is considered that the current and voltage drop drawn into the network are in phase, i.e., $\tan(\varphi) = 0$. In each wind farm, with a particular value of $U$ and $P_r$, the $I_r$ value restricts the cable type that can be used in a connection depending on the number of downstream wind turbines.

To exemplify this, consider the wind farm layout presented in Figure 3, with one substation, node 0, and eight wind turbines, nodes 1 to 8. In this example, assuming that $P_r = 2$ MW and $U = 20$ kV, the rated current drawn by each turbine is $I_r = 57.735$ A (via Equation (8)). This layout has two branch lines: one starting in cable connection $(0,1)$ with blue wind turbines and the other starting in cable connection $(0,2)$ with red wind turbines. The total current reaching the substation from a branch line is the sum of the currents drawn by all turbines connected through this branch. For instance, the branch starting in connection $(0,2)$ supports five wind turbines (including turbine 2). Therefore, the current passing through this cable is $I_{02} = 5 \times I_r = 288.675$ A. Given that, this connection cannot use a cable of type less than five which has $I_z = 313$. Furthermore, in a wind farm with $P_r$ and $U$ values, any branch line cannot have more than $\left\lfloor \frac{I_{z_{10}}}{I_r} \right\rfloor = \left\lfloor \frac{585}{57.7} \right\rfloor = 10$ wind turbines, where $\lfloor a \rfloor$ denotes the maximum integer not greater than $a$.

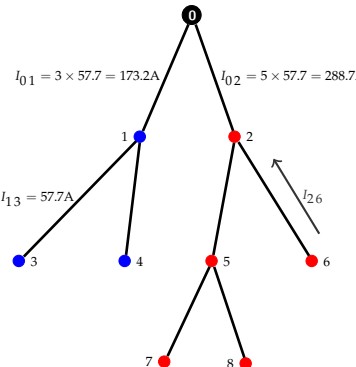

**Figure 3.** Example of a wind farm layout with two main branches (blue WT {1,3,4} and red WT {2,5,6,7,8}). Some branches show the current flowing through them.

## 3. Methodology

The proposed methodology simultaneously combines a genetic algorithm (GA) and integer linear programming (ILP) models, explained in the following. The combination of the two methods enables us to determine the turbines associated with each substation and the connection of the turbines to each substation. The GA, acting at a higher level, is responsible for determining the turbines associated with each substation, and, at a lower level, the ILP model is called upon by the fitness function to determine the optimum link between the turbines and their cable connection types. In this way, the search is performed as a whole.

### 3.1. Genetic Algorithms

Genetic algorithms proposed by Holland [32] were inspired by biology. GAs are widely applied to a wide range of problems. Instances of their use are observed in various fields, including robotics [33], risk management [34], and tunnel lighting [35], among others.

In the GA, the best individuals who can adapt better have greater capacities to reproduce and pass on their genetic material to the next generations. The GA was the first nature-inspired algorithm and is one of the most popular among others. The algorithm is a type of meta-heuristic based on a population of chromosomes, usually called individuals, where each one represents a possible implementation of the problem to be solved. Usually, this representation is achieved through a binary string where a sequence of bits is used to store a problem's parameter.

Initially, the algorithm (see Algorithm 1) begins with a set of chromosomes, called a population, where each one represents a potential solution to the problem. Several cycles are then executed, i.e., generations, where the selection, crossover, and mutation operators are called.

Therefore, selection, crossover, and mutation operators are accountable for searching and optimizing the problem solution. Selection, based on the fitness function, is responsible for guiding the chromosomes over the search space. Selection "randomly" chooses the mates, based on their fitness values. Crossover, or recombination, mimics the natural crossover, where two parents give rise to two offspring, each formed through material from both. Mutation consists of changing the genetic material with a low probability. In the binary implementation, the mutation changes a bit to its complementary.

Search and optimization are then performed through several iterations and usually stopped after a fixed number of iterations. At the end of the algorithm, it is expected that the chromosomes store representations of good ways to solve problems. It should be noted that the GA does not guarantee the optimal solution to the problem but is expected to achieve a good one.

---

**Algorithm 1:** Genetic Algorithm

---

**Result:** Problem solution
t = 0;
fitness(P(t));
random($P(t)$);
**while** $t \leq t_{\max}$ **do**
$\quad$ | $\quad$ P(t+1)=selection(P(t));
$\quad$ | $\quad$ crossover(P(t+1));
$\quad$ | $\quad$ mutation(P(t+1));
$\quad$ | $\quad$ fitness(P(t+1));
$\quad$ | $\quad$ t = t + 1;
**end**

---

### *3.2. Integer Linear Programming*

Integer linear programming (ILP) uses a mathematical model to describe the problem of interest. The adjective linear means that all the mathematical functions in this model are required to be linear functions and the adjective integer indicates that the variables are constrained to have integer values [36]. The word programming is mainly a synonym for planning because linear programming involves the planning of activities to obtain the optimal result, i.e., the one that reaches the specified goal best, among all feasible alternatives. The general form of an integer linear programming model is

$$\min(\max) \quad Z = c_1 x_1 + c_2 x_2 + \cdots + c_n x_n + k \tag{9}$$

$$\text{subject to} \quad a_{11} x_1 + a_{12} x_2 + \cdots + a_{1n} x_n \; \{\leq, \geq, =\} \; b_1 \tag{10}$$

$$\vdots$$

$$a_{m1} x_1 + a_{m2} x_2 + \cdots + a_{mn} x_n \; \{\leq, \geq, =\} \; b_m \tag{11}$$

$$x_j \geq 0, \; j \in \{1, \ldots, n\} \tag{12}$$

$$x_j \in \mathbb{Z}, \; j \in \{1, \ldots, n\} \tag{13}$$

where, $k$, $a_{ij}$, $c_j$, $b_i$, for $i = 1, \ldots, m$ and $j = 1, \ldots, n$, are constants and $x_j$, for $j = 1, \ldots, n$, are the decision variables. The objective function (9) is a linear function on the variables, and it could be to minimize or maximize. Constraints (10) and (11) are the functional constraints and they could be of type "$\leq$", "$\geq$", or "$=$". Finally, constraints (12) and (13) are the domain constraints.

Without the integer constraints, (13), this model is referred to as linear programming (LP). If only some of the variables are required to have integer values, this model is referred to as mixed-integer linear programming (MILP).

## 4. Substation Selection and Cable Layout

In large wind farms, it is common to find several substations. In the design of the cable connections layout, it is necessary to assign turbines to substations. However, if this process is carried out before the cable connections design, the solution obtained via this process could be far from the global optimal. Therefore, it is essential to make these optimizations together.

In this paper, a GA combined with an ILP model is proposed to solve the problem. The GA assigns turbines to a substation, and, during the chromosome evaluation, an ILP optimization model determines its fitness.

The substation assignment and cable connection layout model are described in Sections 4.1 and 4.2, respectively.

### 4.1. Substation Assignment Algorithm

The substation assignment is based on a GA, where the chromosome indicates the substation to which a turbine belongs. Figure 4 presents an example of one chromosome, considering a problem with $|S| = 4$ substations. The value of the gene $j$, or position $j$ of the string, indicates the substation to which the turbine $j$ is assigned. Therefore, the gene is an integer number corresponding to the substation to which turbine $j$ belongs. In the example in Figure 4, turbines 1, 3, 5, 9, and 12 are assigned to substation 1, turbines 2 and 8 are assigned to substation 2, turbines 4 and 10 are assigned to substation 3, and finally, turbines 6, 7, and 11 are assigned to substation 4.

$$\boxed{1\,|\,2\,|\,1\,|\,3\,|\,1\,|\,4\,|\,4\,|\,2\,|\,1\,|\,3\,|\,4\,|\,1}$$

Position:  1  2  3  4  5  6  7  8  9  10  11  12

**Figure 4.** Chromosome representation.

Concerning genetic operators, the selection is based on a tournament between two elements [37], and the crossover operator uses the one-point standard with the probability of $p_c = 0.6$. Figure 5 illustrates the crossover operator at the third cutoff point. In this case, offspring 1 is formed by the first three genes of the first parent and the second parent's last nine genes. The second offspring is formed by the remaining genetic material.

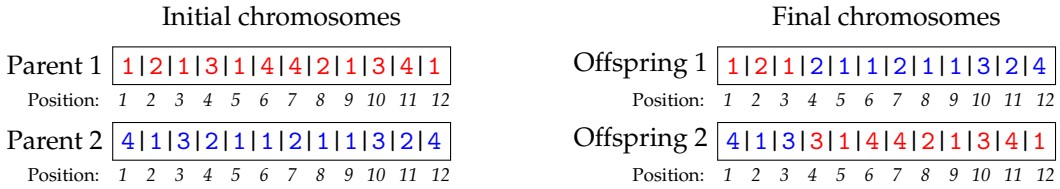

**Figure 5.** Crossover operator.

The mutation operation changes one gene to another possible value, following a uniform distribution. The possible values are $\{1, ..., |S|\}$, where $|S|$ is the number of substations. The probability mutation is $p_m = 0.5/l$, where $l$ is the chromosome length. When a mutation occurs in gene $j$, the assigned substation $0_\ell$ to turbine $j$ is replaced by the substation $k$ with probability:

$$p_m^{jk} = \frac{\frac{1}{d(t_j, 0_k)}}{\sum\limits_{i \in S \setminus \{0_\ell\}} \frac{1}{d(t_j, 0_i)}}. \tag{14}$$

The chromosomes are evaluated through a fitness function (15) that gives a measure of their quality. The sum of several functions gives the fitness. Each of them corresponds to the connection cost between one substation and its turbine set, called a wind field. Given a wind farm where the turbine set is $N$ and the substations set is $S = \{0_1, 0_2, \ldots, 0_{|S|}\}$, the turbine set is partitioned into $|S|$ subsets, $N_i$, each of which will be assigned to the substation $0_i$. The objective function is given by

$$f = \sum_{i=1}^{|S|} f_i(0_i, N_i), \tag{15}$$

where $f_i(0_i, N_i)$ represents the layout cost of assigned turbines set $N_i$ to substation $0_i$, and it will be explained in detail in the next section.

The proposed algorithm's performance was compared with a clustering algorithm. This algorithm connects the turbines (ordinary points) to the nearest substation (centroid). These turbines connect directly or indirectly (passing through other turbines) to the nearest substation. Therefore, in the first phase of this approach, the algorithm determines the

turbines, using the 1-nn algorithm ($k$-nearest neighbor, with $k = 1$), that are connected to each substation. In the second phase, the algorithm runs the proposed ILP for each subset, and the final optimal solution comprises all the optimal solutions from each ILP, with the optimal value being the sum of all the optimal values.

*4.2. Cable Connection Layout Model*

This section presents an integer linear programming (ILP) model to obtain the optimal wind farm layout for a given set of wind turbines and its substation, using the global cost model proposed in [4].

Consider the node set $N = \{1, \ldots, n\}$, corresponding to the wind turbines' locations, and node 0 as the substation location. The goal is to obtain the wind farm connection layout, i.e., a spanning tree of a complete graph with $N_0 = N \cup \{0\}$ as the nodes set, which minimizes the total cost. The total cost is given by the sum of the costs of active losses, $c_p$, and reactive losses, $c_q$, during the expected wind farm lifetime, and the infrastructure cost, $c_I$, which includes the cable costs and their installation. The installation cost corresponds to the digging cost.

For all pairs of nodes $i$ and $j$ (with $i, j \in N_0$), the distance between them is known, and it is denoted by $\ell_{ij}$ (it is assumed that $\ell_{ij} = \ell_{ji}$). Consider the set of available cable types $K = \{1, \ldots, 10\}$ presented in Table 1.

To obtain an ILP model formulation to handle non-linearity between the current intensity, supported by each cable, and its active and reactive losses, as defined by Equations (5) and (6), the decision variables' choice is crucial. Following [4], consider the binary variables $x_{ij}^t$ taking value 1 if the nodes $i$ and $j$ are connected (node $i$ being on the substation side), and this cable connection supports the current of $t$ downstream wind turbines (including the one located in $j$); otherwise, it is zero.

To further clarify, consider the wind farm example presented in Figure 3. The corresponding values of non-null variables are $x_{01}^3 = 1$, $x_{13}^1 = 1$, $x_{14}^1 = 1$, $x_{02}^5 = 1$, $x_{25}^3 = 1$, $x_{26}^1 = 1$, $x_{57}^1 = 1$, and $x_{58}^1 = 1$.

Each cable type $k \in K$ has a maximum current intensity that it can support, $I_{z_k}$, and, therefore, the maximum number of downstream wind turbines for this cable is given by $\left\lfloor \frac{I_{z_k}}{I_r} \right\rfloor$. So, the maximum number of wind turbines in any branch line is:

$$Q = \max_{k \in K} \left\lfloor \frac{I_{z_k}}{I_r} \right\rfloor. \tag{16}$$

Thus, the number of downstream wind turbines that a connection $(i, j)$ can support is:

$$Q(i) = \begin{cases} Q & , i = 0 \\ Q - 1 & , i \in N \end{cases}.$$

The infrastructure cost of making a cable connection between $i$ and $j$ by using a cable of a type $k$ is

$$c_{I ij}^k = (D + 3 \cdot C_k) \cdot \ell_{ij},$$

where $\ell_{ij}$ is the distance between $i$ and $j$, in meters, $D$ is the digging cost, and $3 \cdot C_k$ is the three-phase cable cost (EUR by meter).

By using Equations (5) and (6), the active and reactive losses in a connection $(i, j)$ supporting $t$ downstream turbines, using a three-phase cable of type $k$, are given by:

$$p_{ij}^{kt} = 3 \cdot \frac{\ell_{ji} \cdot R_k}{1000} \cdot (t \cdot I_r)^2 \tag{17}$$

and

$$q_{ij}^{kt} = 3 \cdot \frac{\ell_{ji} \cdot \omega \cdot L_k}{1,000,000} \cdot (t \cdot I_r)^2, \tag{18}$$

where $\omega$ is the angular frequency. Therefore, the costs of the active and reactive losses in a connection $(i, j)$ supporting $t$ downstream turbines, using a three-phase cable of type $k$, during the wind farm lifetime, are, respectively,

$$c_{p_{ij}}^{kt} = 3 \cdot h \cdot c_e \frac{\ell_{ji} \cdot R_k}{1000} \cdot t^2 \cdot l_f^2 \cdot I_r^2, \tag{19}$$

and

$$c_{q_{ij}}^{kt} = 3 \cdot h \cdot 0.5 \, c_e \frac{\ell_{ji} \cdot \omega \cdot L_k}{1,000,000} \cdot t^2 \cdot l_f^2 \cdot I_r^2 \tag{20}$$

where $h$ is the number of hours during the expected wind farm lifetime, $c_e$ is the energy cost, and $l_f$ is the load factor. The value of $l_f$ reflects the real operating conditions during the wind farm lifetime and is the ratio between the generated current and the maximum current that can be generated. Furthermore, according to the Portuguese market price, the cost of reactive losses is half that of the active losses, and so in the expression of the reactive losses costs, (20), it appears $0.5 \, c_e$.

The optimal cable type for a connection $(i, j)$ supporting the current of $t$ downstream wind turbines does not depend on the other links, and so, it can be computed previously. Following [4], it is merely the type $k$ that minimizes the sum $c_{I_{ij}}^k + c_{p_{ij}}^{kt} + c_{q_{ij}}^{kt}$, assuring that the current intensity generated by $t$ wind turbines is not higher than the current intensity $I_{z_k}$, i.e., $I_{z_k} \geq t \cdot I_r$. Therefore, for each $(i, j)$ with $i \in N_0$ and $j \in N$ and $t \in \{0, \ldots, Q(i)\}$, the optimal cable type is given by:

$$k_{ij}^t = \arg \min_{k \in K: t \cdot I_r \leq I_{z_k}} \left( c_{I_{ij}}^k + c_{p_{ij}}^{kt} + c_{q_{ij}}^{kt} \right), \tag{21}$$

and the corresponding cost, for cable type $k = k_{ij}^t$,

$$L_{ij}^t = c_{I_{ij}}^k + c_{p_{ij}}^{kt} + c_{q_{ij}}^{kt} \tag{22}$$

is the minimum cost for the connection $(i, j)$ with $t$ downstream turbines, including the infrastructure cost and both types of losses costs.

The ILP model, Layout, is given by:

$$\min \quad \sum_{i \in N_0} \sum_{j \in N} \sum_{t=1}^{Q(i)} L_{ij}^t \cdot x_{ij}^t \tag{23}$$

subject to

$$\sum_{j \in N} \sum_{t=1}^{Q} \left( t \cdot x_{0j}^t \right) = n \tag{24}$$

$$\sum_{i \in N_0} \sum_{t=1}^{Q(i)} x_{ij}^t = 1, j \in N \tag{25}$$

$$\sum_{i \in N_0} \sum_{t=1}^{Q(i)} \left( t \cdot x_{ij}^t \right) = \sum_{i \in N} \sum_{t=1}^{Q(i)} \left( t \cdot x_{ji}^t \right) + 1, j \in N \tag{26}$$

$$x_{ij}^t \in \{0, 1\}, i \in N_0, j \in N, t = 1, \ldots, Q(i) \tag{27}$$

The objective function (23) aims to minimize the sum of infrastructure costs and both types of losses costs. Constraint (24) guarantees that the network connects all wind turbines. Constraints (25) assure that each wind turbine $j \in N$ has one incoming connection. Constraints (26) are the flow conservation constraints and guarantee that, for each wind turbine $j \in N$, if there exists an incoming connection supporting $t$ downstream wind turbines (left-hand side of the constraints), then the outgoing connections from this turbine

must support a total of $t - 1$ downstream wind turbines (right-hand side of the constraints). Finally, constraints (27) are the variable domain constraints.

This model was proposed in [4], and it is close to the model proposed for the Capacitated Minimum Spanning Tree (CMST) problem in [38].

It is well known that the performance of exact algorithms based on mathematical models, such as the branch-and-bound and the branch-and-cut, depend greatly on the quality of the model; see [39] for integer programming problems in general. To improve the model, the inclusion of valid inequalities is a very widely used technique. A valid inequality for an ILP is any constraint that does not eliminate any feasible integer solution, but its inclusion tightens the formulation, leading to a formulation whose linear relaxation is closer to the convex hull of the set of feasible solutions.

Following [4], a set of valid inequalities able to improve the models' efficiency is used as a set of additional constraints to improve the model Layout:

$$\sum_{i \in N} \sum_{\tau=t}^{Q(j)} \left( x_{ji}^{\tau} \right) \leq \sum_{i \in N_0} \sum_{\tau=t+1}^{Q(i)} \left( \left\lfloor \frac{\tau - 1}{t} \right\rfloor \cdot x_{ij}^{\tau} \right), j \in N, t = 2, \ldots, Q - 2 \tag{28}$$

$$\sum_{i \in N} \left( x_{ji}^{Q(j)} \right) \leq x_{0j}^{Q}, j \in N \tag{29}$$

These inequalities are based on the fact that if a connection going into node $j \in N$ supports $\tau$ downstream wind turbines, then the number of connections outgoing from this node supporting at least $t$ downstream wind turbines cannot be higher than $\left\lfloor \frac{\tau-1}{t} \right\rfloor$.

The ILP model that adds constraints (28) and (29) to the Layout model is designated as the Layout+ model.

## 5. Results and Discussion of the Case Studies

This section presents and discusses the results using the proposed methodology. The GA was written in Python, and the fitness function requires solving the integer programming model Layout+. The integer programming models are solved using the Xpress optimizer library. Table 2 sums up the GA parameters used, which were chosen based on computational experiments.

**Table 2.** Genetic algorithm parameters.

| Parameter | Value |
| --- | --- |
| Chromosome length, $l$ | Number of turbines |
| Selection | Tournament of two |
| Crossover probability, $p_c$ | 0.6 |
| Mutation probability, $p_m$ | $\frac{0.5}{l}$ |
| Population | 100 |
| Generations | 500 |

The proposed methodology was applied to four wind farms, with two, three, four, and five substations. The first wind farm is the *Alto da Coutada* wind farm, located in the north of Portugal, formed of 79 turbines and two substations. The second wind farm, called WF-S3, is formed of 74 turbines and three substations, generated according to an example in [40]. The third wind farm, called WF-S4, has four substations and 79 turbines. The last wind farm is the *Alto Minho*, located in the north of Portugal, formed of 120 turbines and five substations.

In all case studies, the following were considered: ten cable types presented in Table 1, $c_e = 102.52 \times 10^{-6}$ EUR/Wh as the energy cost, $h = 24 \times 365 \times 20$ as the number of hours during the expected lifetime of the wind farm (20 years ), $l_f = 0.5$ as the load factor, and $\omega = 100\pi$ rad/s as the angular frequency.

### 5.1. Alto da Coutada Wind Farm

The first case study is the Alto da Coutada wind farm with two substations and 50 wind turbines with $P_r = 2$ MW of rated power, interconnected by a $U = 20$ kV grid. With these parameters, the rated current drawn by each turbine is $I_r = 57.735$ A, and the maximum number of wind turbines per branch line is $Q = 10$. The coordinates of the wind turbines are in [4], and the coordinates of the two substations are $\{(41.5227550, -7.5958050), (41.5792210, -7.5358600)\}$.

The obtained cable connection layout is presented in Figure 6. The wind farm has two wind fields (a parcel of a wind farm with one substation), $(0_1, N_1)$ and $(0_2, N_2)$, where the turbine sets linked to substations $0_1$ and $0_2$ are, respectively, $N_1 = \{1, 2, 3, 4, 5, 6, 7, 8, 9, 10, 11, 12, 13, 14, 15\}$ and $N_2 = \{16, 17, 18, 19, 20, 21, 22, 23, 24, 25, 26, 27, 28, 29, 30, 31, 32, 33, 34, 35, 36, 37, 38, 39, 40, 41, 42, 43, 44, 45, 46, 47, 48, 49, 50\}$. On each wind field, the turbines are filled with the same color.

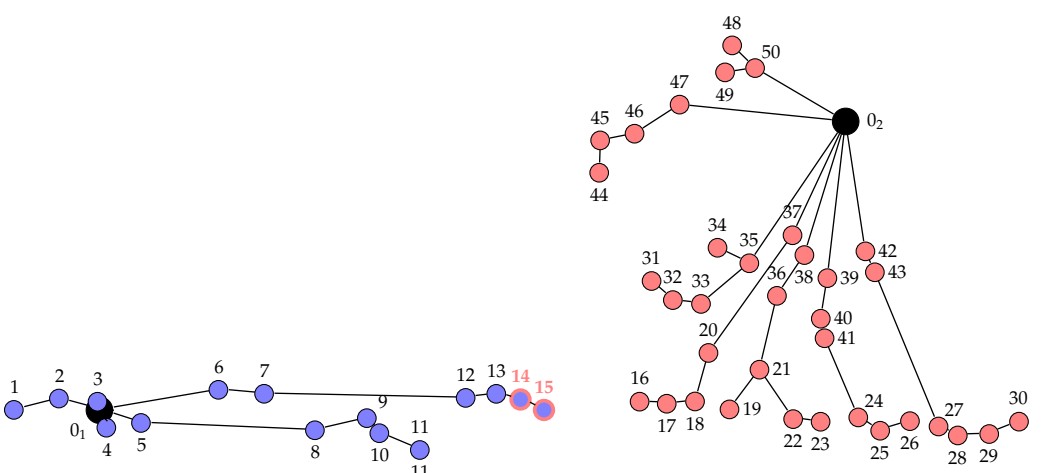

**Figure 6.** Turbines and cable connection layout for the *Alto da Coutada* wind farm. In blue are the WTs connected to Substation $O_1$ and in orange are the WTs connected to Substation $O_2$. In blue with orange borders are the WTs that would be connected to Substation $O_2$ using the clustering method.

Table 3 shows the connections and their cable type for each wind field in the final solution and the corresponding costs. The first column presents the substation, $0_i$, and the number of turbines assigned to it, $|N_i|$. The next three columns, "$k$", "Links", and "#Links", indicate the type of cables used, the links $(i, j)$ of this cable type, and the corresponding number of cables, in the solution. Finally, the column "Cost" presents the global total cost $f$ of the wind farm, the total cost $f_i(0_i, N_i)$ of the wind fields $(0_i, N_i)$, and the partial cost of each wind field, namely, infrastructure cost $c_I$, active losses $c_p$, and reactive losses $c_q$.

The total cost layout of the two wind fields is EUR 4,795,930.7. The wind field $(0_1, N_1)$ has 15 wind turbines, and the total cost is EUR 1,239,980.2, where 55.7% is the infrastructure cost, corresponding to EUR 690,802.4, 24.1% is the active losses cost, corresponding to EUR 298,316.2, and 20.2% is the reactive losses cost, corresponding to EUR 25,0861.6. The wind field $(0_2, N_2)$ has 35 wind turbines, and its total cost is EUR 3,555,951.7, where 54.7% is the infrastructure cost, 23.9% is the active losses cost, and the remaining 21.4% is the reactive losses cost. The highest amount corresponds to the infrastructure cost, and the smallest cost is the reactive losses cost during the wind farm lifetime.

The clustering algorithm is used to infer the GA's performance. The turbines are first assigned to the closest substation via the clustering algorithm, followed by the ILP model, Layout+, to optimize the wind farm layout considering the WT assignments. The obtained solution is then compared with the one obtained in the previous section. The turbine sets assigned to substations $0_1$ and $0_2$ are, respectively, $N_1 = \{1, 2, 3, 4, 5, 6, 7, 8, 9, 10, 11, 12, 13\}$ and $N_2 = \{14, 15, 16, 17, 18, 19, 20, 21, 22, 23, 24, 25, 26, 27, 28, 29, 30, 31, 32, 33, 34, 35, 36, 37, 38, 39, 40, 41, 42, 43, 44, 45, 46, 47, 48, 49, 50\}$. The obtained objective value corresponds

to the cost EUR 4,800,839.0. This value is higher than the one obtained via the GA. It can be observed that, in the solution presented in Section 5.1, turbines 14 and 15 are not assigned to the nearest substation, $0_2$. Thus, it can be concluded that the clustering algorithm could lead to a worse solution than the one obtained with the proposed methodology. In Figure 6, turbines grouped differently in the two algorithms are highlighted using different colors for the labels. Turbines with different label colors mean that the turbine is connected to another substation (according to its color) in the clustering algorithm.

**Table 3.** Solution description for *Alto da Coutada* wind farm.

| Wind Field | $k$ | Links | #Links | Cost (EUR) |
|---|---|---|---|---|
| $0_1$ | 3 | $(0_1, 3)$, $(0_1, 4)$, $(2, 1)$, $(10, 11)$, $(14, 15)$ | 5 | $f_1(0_1, N_1) = 1{,}239{,}980.2$ |
| | 4 | $(0_1, 2)$, $(9, 10)$, $(13, 14)$ | 3 | $c_I(0_1, N_1) = 690{,}802.4$ |
| $\|N_1\| = 15$ | 7 | $(8, 9)$, $(12, 13)$ | 2 | $c_p(0_1, N_1) = 298{,}316.2$ |
| | 8 | $(5, 8)$, $(7, 12)$ | 2 | $c_q(0_1, N_1) = 250{,}861.6$ |
| | 10 | $(0_1, 5)$, $(0_1, 6)$, $(6, 7)$ | 3 | |
| $0_2$ | 3 | $(17, 16)$, $(21, 19)$, $(22, 23)$, $(25, 26)$, $(29, 30)$, $(32, 31)$, $(35, 34)$, $(45, 44)$, $(50, 48)$, $(50, 49)$ | 10 | $f_2(0_2, N_2) = 3{,}555{,}950.5$ |
| | 4 | $(18, 17)$, $(21, 22)$, $(24, 25)$, $(28, 29)$, $(33, 32)$, $(46, 45)$ | 6 | $c_I(0_2, N_2) = 1{,}944{,}381.7$ |
| $\|N_2\| = 35$ | 7 | $(0_2, 50)$, $(20, 18)$, $(27, 28)$, $(35, 33)$, $(41, 24)$, $(47, 46)$ | 6 | $c_p(0_2, N_2) = 852{,}218.4$ |
| | 8 | $(0_2, 47)$, $(36, 21)$, $(37, 20)$, $(40, 41)$, $(43, 27)$ | 5 | $c_q(0_2, N_2) = 759{,}350.4$ |
| | 10 | $(0_2, 35)$, $(0_2, 37)$, $(0_2, 38)$, $(0_2, 39)$, $(0_2, 42)$, $(38, 36)$, $(39, 40)$, $(42, 43)$ | 8 | |
| | | | | $f = 4{,}795{,}930.7$ |

### 5.2. WF-S3 Wind Farm

The second case study is the WF-S3 wind farm with three substations and 74 wind turbines with $P_r = 2$ MW of rated power, interconnected by a $U = 20$ kV grid. With these parameters, the rated current drawn by each turbine is $I_r = 57.735$ A, and the maximum number of wind turbines per branch line is $Q = 10$. The coordinates of the wind turbines and substations are in Table A1.

The obtained cable connection layout is presented in Figure 7. The wind farm has three wind fields, $(0_1, N_1)$, $(0_2, N_2)$, and $(0_3, N_3)$, where the turbine sets linked to substations $0_1$, $0_2$, and $0_3$ are, respectively, $N_1 = \{1, 2, 3, 4, 5, 6, 7, 8, 9, 10, 14, 17, 19, 20, 21, 22, 23, 24\}$, $N_2 = \{18, 25, 26, 27, 28, 29, 30, 31, 32, 33, 34, 35, 36, 37, 38, 39, 40, 41, 42, 43, 44, 45, 46, 47, 48, 49\}$, and $N_3 = \{11, 12, 13, 15, 16, 50, 51, 52, 53, 54, 55, 56, 57, 58, 59, 60, 61, 62, 63, 64, 65, 66, 67, 68, 69, 70, 71, 72, 73, 74\}$. In the presented solution, it can be observed that some border turbines are not assigned to the nearest substation.

Table 4 characterizes the connections and costs for each wind field in the final solution. The total cost layout of the three wind fields is EUR 2,838,121.1. The wind field $(0_1, N_1)$ has 18 wind turbines, and the total cost is EUR 658,709.8, where 70.4% is the infrastructure cost, corresponding to EUR 463,373.1, 19.2% is the active losses cost, corresponding to EUR 126,267.6, and 10.5% is the reactive losses cost, corresponding to EUR 69,069.1. The wind field $(0_2, N_2)$ has 26 wind turbines. Its total cost is EUR 1,036,720.7, where 64.0% is the infrastructure cost, 22.6% is the active losses cost, and the remaining 13.4% is the reactive losses cost. The wind field $(0_3, N_3)$ has 30 wind turbines, and the total cost is EUR 1,142,690.6, where 64.9% is the infrastructure cost, 22.8% is the active losses cost, and the remaining 12.3% is the reactive losses cost. Again, the higher cost is the infrastructure and the lower cost is the cost of reactive losses over the lifetime of the wind farm.

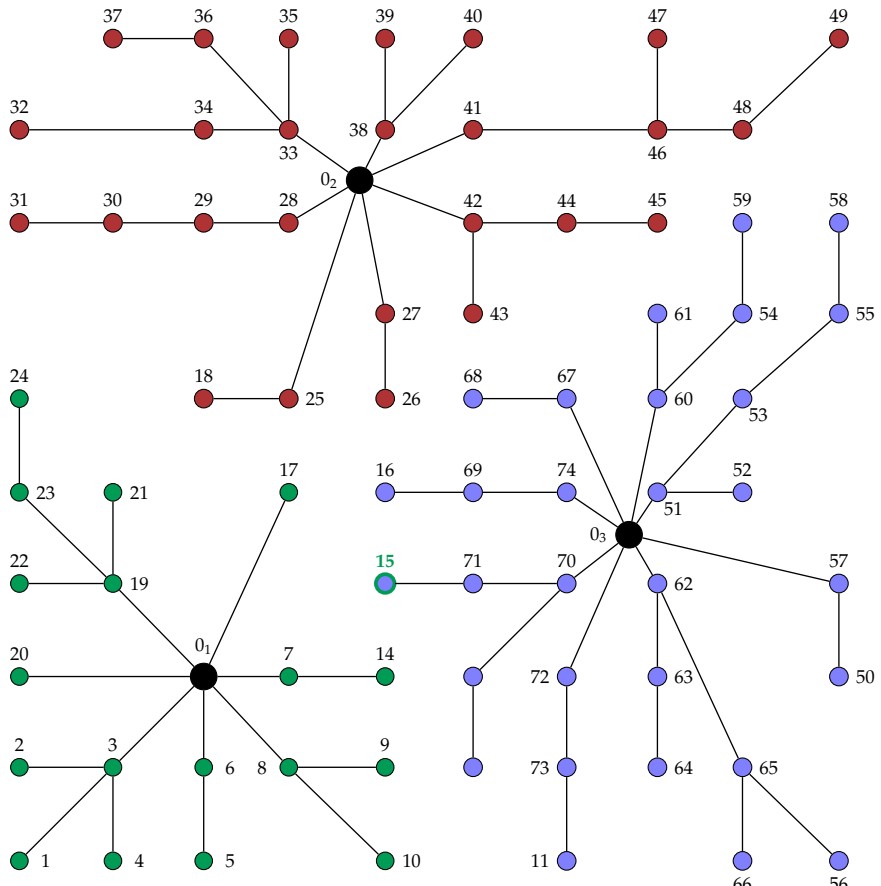

**Figure 7.** Turbines and the cable connection layout for the WF-3S wind farm. The WTs connected to substations $O_1$, $O_2$, and $O_3$ are shown in green, red, and blue, respectively. In blue with a green border are the WTs that would be connected to Substation $O_1$ using the clustering method.

**Table 4.** Solution description of WF-S3 wind farm.

| Wind Field | $k$ | Links | #Links | Cost (EUR) |
|---|---|---|---|---|
| $0_1$ | 3 | $(0_1, 17)$, $(0_1, 20)$, $(3, 1)$, $(3, 2)$, $(3, 4)$, $(6, 5)$, $(7, 14)$, $(8, 9)$, $(8, 10)$, $(19, 21)$, $(19, 22)$, $(23, 24)$ | 12 | $f_1(0_1, N_1) = 658{,}709.8$ |
| $\|N_1\| = 18$ | 4 | $(0_1, 6)$, $(0_1, 7)$, $(19, 23)$ | 3 | $c_I(0_1, N_1) = 463{,}373.1$ |
| | 7 | $(0_1, 8)$ | 1 | $c_p(0_1, N_1) = 126{,}267.6$ |
| | 8 | $(0_1, 3)$ | 1 | $c_q(0_1, N_1) = 69{,}069.1$ |
| | 10 | $(0_1, 19)$ | 1 | |
| $0_2$ | 3 | $(25, 18)$, $(27, 26)$, $(30, 31)$, $(33, 35)$, $(34, 32)$, $(36, 37)$, $(38, 39)$, $(38, 40)$, $(42, 43)$, $(44, 45)$, $(46, 47)$, $(48, 49)$ | 12 | $f_2(0_1, N_2) = 1{,}036{,}720.6$ |
| $\|N_2\| = 26$ | 4 | $(0_2, 25)$, $(0_2, 27)$, $(29, 30)$, $(33, 34)$, $(33, 36)$, $(42, 44)$, $(46, 48)$ | 7 | $c_I(0_2, N_2) = 663{,}759.7$ |
| | 7 | $(0_2, 38)$, $(28, 29)$ | 2 | $c_p(0_2, N_2) = 234{,}065.8$ |
| | 8 | $(00, 28)$, $(0_2, 42)$, $(41, 46)$ | 3 | $c_q(0_2, N_2) = 138{,}895.2$ |
| | 10 | $(0_2, 33)$, $(0_2, 41)$ | 2 | |
| $0_3$ | 3 | $(13, 12)$, $(51, 52)$, $(54, 59)$, $(55, 58)$, $(57, 50)$, $(60, 61)$, $(63, 64)$, $(65, 56)$, $(65, 66)$, $(67, 68)$, $(69, 16)$, $(71, 15)$, $(73, 11)$ | 13 | $f_3(0_3, N_3) = 1{,}142{,}690.6$ |
| $\|N_3\| = 30$ | 4 | $(0_3, 57)$, $(0_3, 67)$, $(53, 55)$, $(60, 54)$, $(62, 63)$, $(70, 13)$, $(70, 71)$, $(72, 73)$, $(74, 69)$ | 9 | $c_I(0_3, N_3) = 741{,}715.7$ |
| | 7 | $(0_3, 72)$, $(0_3, 74)$, $(51, 53)$, $(62, 65)$ | 4 | $c_p(0_3, N_3) = 261{,}035.2$ |
| | 8 | $(0_3, 60)$ | 1 | $c_q(0_3, N_3) = 139{,}939.7$ |
| | 10 | $(0_3, 51)$, $(0_3, 62)$, $(0_3, 70)$ | 3 | |
| | | | | $f = 2{,}838{,}121.1$ |

To complete the proposed approach's performance analysis, we will present the solution using a clustering algorithm to assign turbines to the closest substation, instead of the GA, followed by the ILP model, Layout+. The turbine sets assigned to substations $0_1$, $0_2$, and $0_3$ are, respectively, $N_1 = \{1, 2, 3, 4, 5, 6, 7, 8, 9, 10, 14, 15, 17, 19, 20, 21, 22, 23, 24\}$, $N_2 = \{18, 25, 26, 27, 28, 29, 30, 31, 32, 33, 34, 35, 36, 37, 38, 39, 40, 41, 42, 43, 44, 45, 46, 47, 48, 49\}$, and $N_3 = \{11, 12, 13, 16, 50, 51, 52, 53, 54, 55, 56, 57, 58, 59, 60, 61, 62, 63, 64, 65, 66, 67, 68, 69, 70, 71, 72, 73, 74\}$. The obtained objective value corresponds to the cost EUR 2,839,945.3. This value is higher than the one obtained via the GA. Indeed, for turbine 15, although it is closer to substation $0_1$, the connection of it to substation $0_3$ leads to a cheaper solution.

*5.3. WF-S4 Wind Farm*

The third case study is the WF-S4 wind farm with four substations and 79 wind turbines with $P_r = 2$ MW of rated power, interconnected by a $U = 20$ kV grid. With these parameters, the rated current drawn by each turbine is $I_r = 57.735$ A, and the maximum number of wind turbines per branch line is $Q = 10$. The coordinates of the wind turbines and substations are in Table A2.

The obtained cable connection layout is presented in Figure 8. The wind farm has four wind fields, $(0_1, N_1)$, $(0_2, N_2)$, $(0_3, N_3)$, and $(0_4, N_4)$, where the turbine sets linked to substations $0_1$, $0_2$, $0_3$, and $0_4$ are, respectively, $N_1 = \{3, 4, 5, 6, 7, 13, 14, 15, 16, 22, 23, 24, 32, 33, 34\}$, $N_2 = \{40, 44, 45, 46, 50, 51, 52, 57, 58, 59, 60, 61, 62, 68, 69, 70, 71, 72, 73, 77\}$, $N_3 = \{1, 2, 11, 12, 21, 26, 27, 28, 29, 30, 31, 38, 39, 42, 43, 49, 55, 56, 65, 66, 67, 75, 76\}$, and $N_4 = \{8, 9, 10, 17, 18, 19, 20, 25, 35, 36, 37, 41, 47, 48, 53, 54, 63, 64, 74, 78, 79\}$.

Table 5 shows the connections and cable type presented in the final solution and the corresponding costs for each wind field.

**Table 5.** Solution description for the WF-S4 wind farm.

| Wind Field | $k$ | Links | #Links | Cost (EUR) |
|---|---|---|---|---|
| $0_1$ | 3 | $(4, 3)$, $(7, 16)$, $(13, 22)$, $(14, 33)$, $(15, 24)$, $(33, 32)$ | 6 | $f_1(0_1, N_1) = 1{,}217{,}848.1$ |
| | 4 | $(4, 13)$, $(15, 25)$, $(24, 34)$ | 3 | $c_I(0_1, N_1) = 771{,}301.4$ |
| $|N_1| = 15$ | 7 | $(0_1, 14)$, $(6, 7)$ | 2 | $c_p(0_1, N_1) = 289{,}847.9$ |
| | 8 | $(5, 8)$, $(7, 12)$ | 2 | $c_q(0_1, N_1) = 156{,}698.7$ |
| | 10 | $(0_1, 6)$, $(0_1, 15)$, $(5, 4)$ | 3 | |
| $0_2$ | 3 | $(0_2, 77)$, $(44, 40)$, $(45, 46)$, $(51, 44)$, $(59, 50)$, $(59, 58)$, $(70, 71)$ | 7 | $f_2(0_2, N_2) = 1{,}760{,}766.6$ |
| | 4 | $(52, 45)$, $(58, 57)$, $(60, 51)$, $(68, 59)$ | 4 | $c_I(0_2, N_2) = 1{,}089{,}737.5$ |
| $|N_2| = 20$ | 7 | $(69, 60)$ | 1 | $c_p(0_2, N_2) = 392{,}414.2$ |
| | 8 | $(69, 60)$, $(71, 62)$ | 2 | $c_q(0_2, N_2) = 278{,}614.9$ |
| | 10 | $(0_2, 52)$, $(0_2, 68)$, $(0_2, 69)$, $(0_2, 70)$ | 4 | |
| $0_3$ | 3 | $(0_3, 65)$, $(11, 2)$, $(11, 12)$, $(29, 30)$, $(30, 31)$, $(43, 38)$ | 6 | $f_3(0_3, N_3) = 2{,}080{,}631.6$ |
| | 4 | $(21, 1)$, $(28, 29)$, $(42, 43)$ | 3 | $c_I(0_3, N_3) = 1{,}297{,}036.9$ |
| $|N_3| = 23$ | 7 | $(0_3, 26)$, $(0_3, 55)$, $(26, 21)$, $(27, 11)$, $(38, 39)$ | 5 | $c_p(0_3, N_3) = 501{,}304.0$ |
| | 8 | $(0_3, 27)$, $(0_3, 28)$, $(0_3, 42)$, $(0_3, 49)$ | 4 | $c_q(0_3, N_3) = 282{,}290.6$ |
| $0_4$ | 3 | $(9, 8)$, $(18, 17)$, $(20, 10)$, $(36, 35)$, $(48, 47)$, $(48, 63)$ | 6 | $f_4(0_4, N_4) = 2{,}099{,}821.7$ |
| | 4 | $(0_4, 20)$, $(19, 9)$, $(37, 25)$, $(41, 37)$, $(54, 53)$ | 5 | $c_I(0_4, N_4) = 1{,}297{,}584.1$ |
| $|N_4| = 21$ | 7 | $(37, 36)$, $(47, 54)$ | 2 | $c_p(0_4, N_4) = 466{,}272.4$ |
| | 8 | $(0_4, 41)$, $(0_4, 64)$ | 2 | $c_q(0_4, N_4) = 335{,}965.2$ |
| | 10 | $(0_4, 19)$, $(0_4, 48)$, $(19, 18)$ | 3 | |
| | | | | $f = 7{,}159{,}067.8$ |

The total cost layout of the four wind fields is EUR 7,159,067.8. The wind field $(0_1, N_1)$ has 15 wind turbines, and the total cost is EUR 1,217,848.0, where 63.3% is the infrastructure cost, corresponding to EUR 771,301.4, 23.8% is the active losses cost, corresponding to EUR 289,847.9, and 12.9% is the reactive losses cost, corresponding to EUR 156,698.7. The wind field $(0_2, N_2)$ has 20 wind turbines, and the total cost is EUR 1,760,766.6, where 61.9%

is the infrastructure cost, 22.3% is the active losses cost, and the remaining 15.8% is the reactive losses cost. Wind field $(0_3, N_3)$ has 23 wind turbines, and the total cost is EUR 2,080,631.6, where 62.3% is the infrastructure cost, 24.1% is the active losses cost, and 13.6% is the reactive losses cost. The wind field $(0_4, N_4)$ has 21 wind turbines, and the total cost is EUR 2,099,821.7, with EUR 1,297,584.1 for the infrastructure cost, EUR 466,272.4 for the active losses cost, and EUR 335,965.2 for the reactive losses cost.

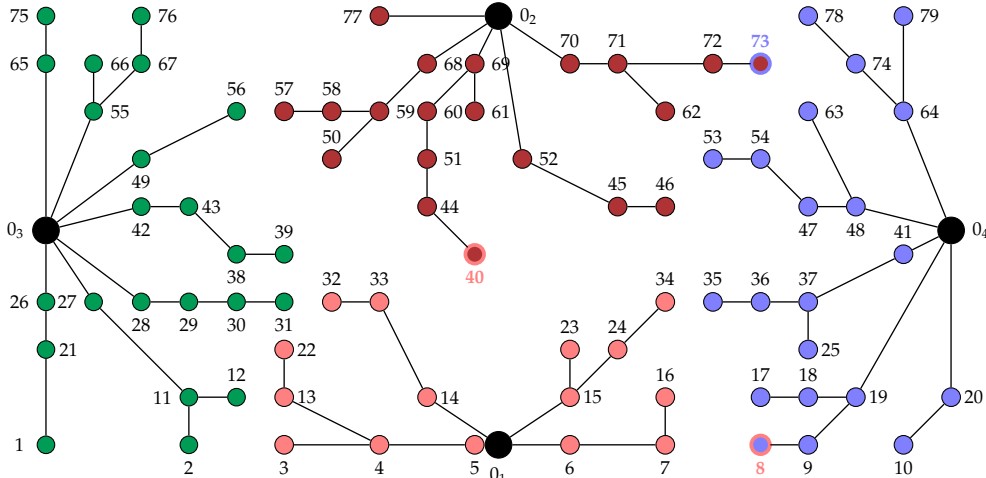

**Figure 8.** Turbines and cable connection layout for the WF-S4 wind farm. The WTs connected to substations $O_1$, $O_2$, $O_3$, and $O_4$ are shown in orange, red, green, and blue, respectively. Using the clustering method, WTs 8 and 40 should be connected to $O_1$, and WT 73 should be connected to $O_4$.

The WT-S4 wind farm results where the turbines were first assigned to the closest substation via a clustering algorithm followed by the ILP model Layout+ are presented in the following. The turbine sets assigned to substations $0_1$, $0_2$, $0_3$, and $0_4$ are, respectively, $N_1 = \{3, 4, 5, 6, 7, 8, 13, 14, 15, 16, 22, 23, 24, 32, 33, 34, 40\}$, $N_2 = \{44, 45, 46, 50, 51, 52, 57, 58, 59, 60, 61, 62, 68, 69, 70, 71, 72, 77\}$, $N_3 = \{1, 2, 11, 12, 21, 26, 27, 28, 29, 30, 31, 38, 39, 42, 43, 49, 55, 56, 65, 66, 67, 75, 76\}$, and $N_4 = \{9, 10, 17, 18, 19, 20, 25, 35, 36, 37, 41, 47, 48, 53, 54, 63, 64, 73, 74, 78, 79\}$. The obtained objective value corresponds to the cost EUR 7,170,952.23. This value is higher than the one obtained via the GA. Although turbines 8 and 40 are closer to substation $0_1$ and turbine 73 is closer to substation $0_4$, a cheaper solution is obtained if they are connected to substations $0_4$ and $0_2$, respectively.

*5.4. Alto Minho Wind Farm*

The last case study is the *Alto Minho* wind farm, with five substations and 120 turbines. This wind farm spans four locations, namely, *Picos*, *São Silvestre*, *Santo António*, and *Mendoiro*. *Santo António* has two substations and the others have one each. *Picos* has 26 turbines, *São Silvestre* has 19 turbines, *Santo António* has 61 and 33 (94), and *Mendoiro* 26. So, the park is divided into five well-defined regions. Therefore, it is easy to determine the turbine set assigned to each substation. However, the GA is used to solve the problem to validate the algorithm. The results obtained are depicted in Figure 9, where each wind field $(0_i, N_i)$ is illustrated with different colors. Figure 9a shows all the regions, the black circles point out the substations, and the other colored circles indicate the turbines. It can be easily seen that the GA divides the turbines well over the fields. Figure 9b zooms in on the *Picos* wind field, and Figure 9c zooms in on the *São Silvestre* wind field, Figure 9d zooms in on the *Mendoiro* wind field. Figure 9e zooms in on the *Santo António* I wind field, and finally, Figure 9f zooms in on the *Santo António* II wind field.

The turbine sets linked to substations $0_1$, $0_2$, $0_3$, $0_4$, and $0_5$ are, respectively, $N_1 = \{1, 2, 3, 4, 5, 6, 7, 8, 9, 10, 12, 13, 14, 18, 19, 20, 21, 22, 23, 24, 25, 26\}$, $N_2 = \{27, 28, 29, 30, 31, 32, 33, 34, 35, 36, 37, 38, 39, 40, 41, 42, 43, 44, 45\}$, $N_3 = \{46, 47, 48, 49, 50, 51, 52, 53, 54, 55, 56, 57, 58, 59, 60, 61\}$, $N_4 = \{62, 63, 64, 65, 66, 67, 68, 69, 70, 71, 72, 73, 74, 75, 76, 77, 78, 79, 80, 81, 82,$

83, 84, 85, 86, 87, 88, 89, 90, 91, 92, 93, 94}, and $N_5$ = {95, 96, 97, 98, 99, 100, 101, 102, 103, 104, 105, 106, 107, 108, 109, 110, 111, 112, 113, 114, 115, 116, 117, 118, 119, 120}.

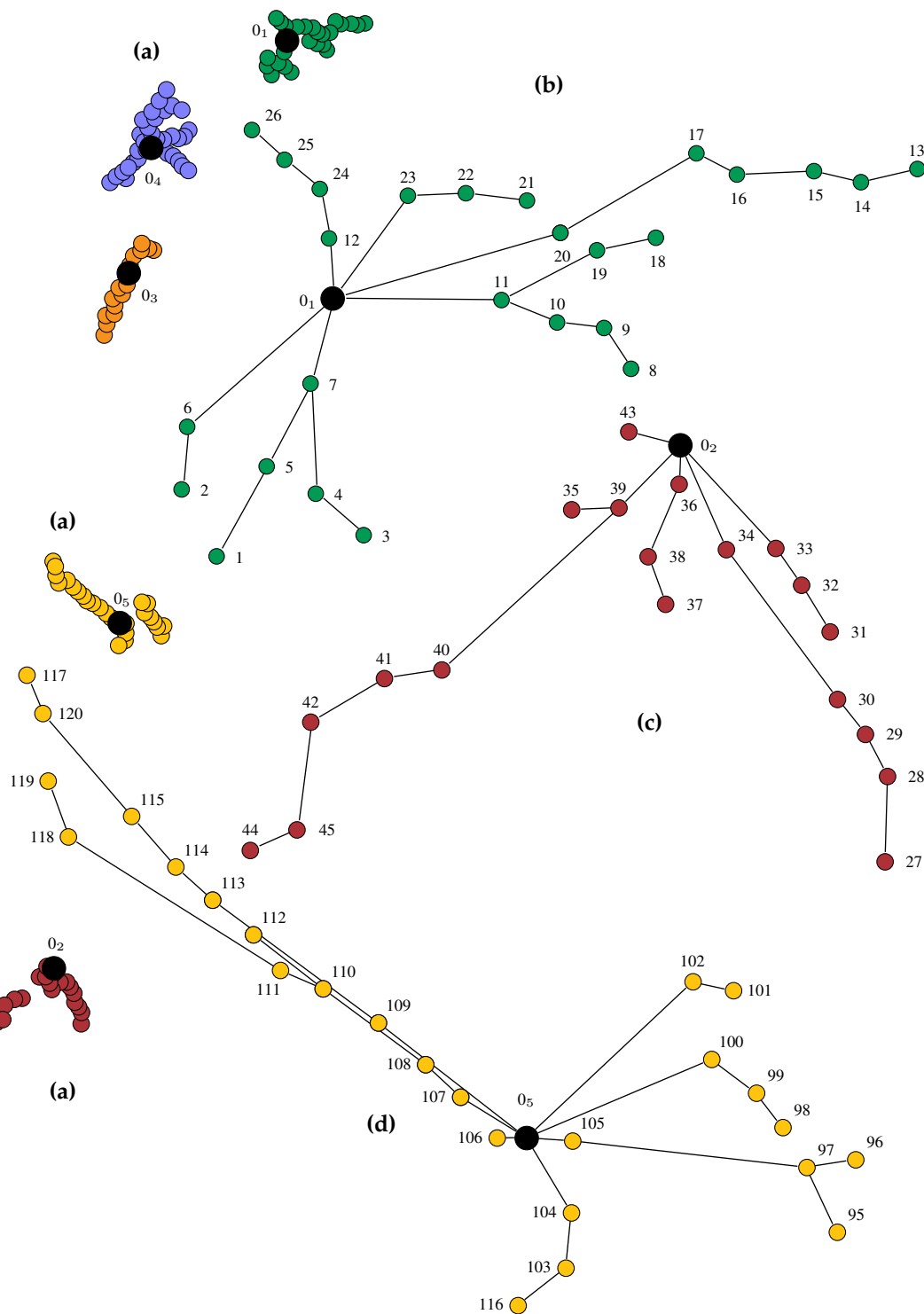

**Figure 9.** *Cont.*

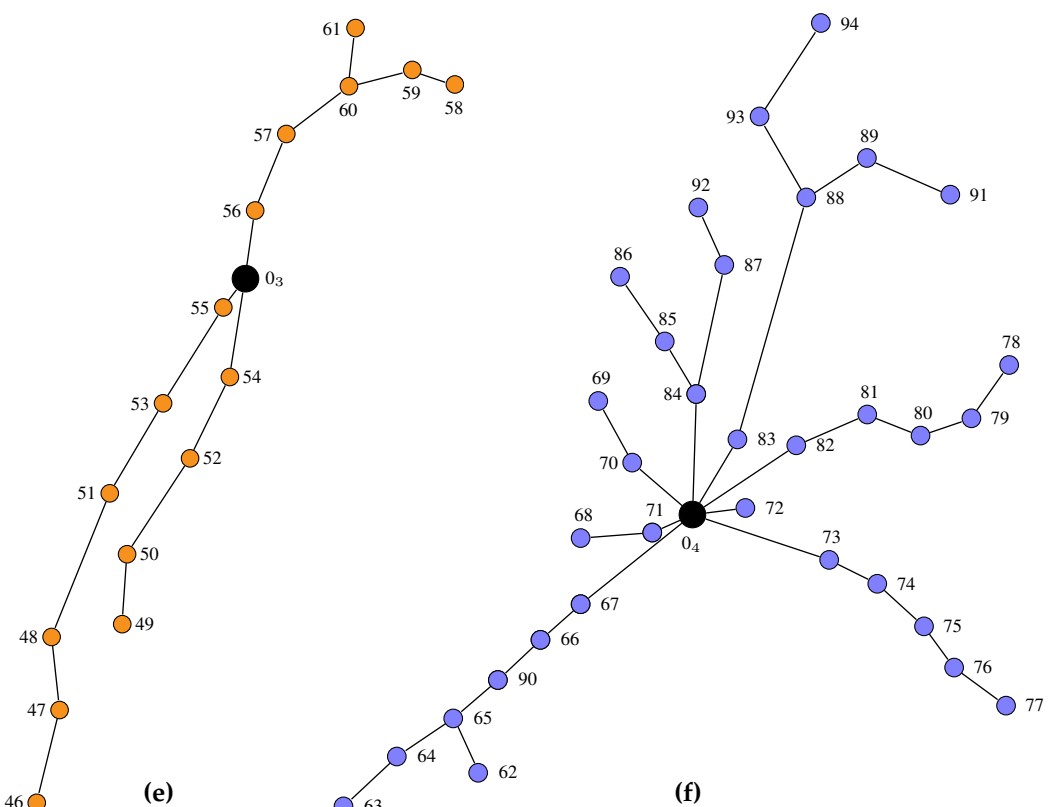

**Figure 9.** Turbines and cable connection layout for the *Alto Minho* wind farm of the (**a**) entire park, (**b**) *Picos* wind field, (**c**) *São Silvestre* wind field, (**d**) *Mendoiro* wind field, (**e**) *Santo António* I wind field, and (**f**) *Santo António* II wind field.

Table 6 shows the connections and cable type presented in the final solution and the corresponding costs for each wind field.

The total cost layout of the five wind fields is EUR 5,439,809.2. The wind field $(0_1, N_1)$ has 26 wind turbines, and the total cost is EUR 1,255,170.5, where 57.0% is the infrastructure cost, corresponding to EUR 715,716.4, 23.6% is the active losses cost, corresponding to EUR 296,430.9, and 19.4% is the reactive losses cost, corresponding to EUR 243,023.2. The wind field $(0_2, N_2)$ has 19 wind turbines, and the total cost is EUR 873,066.6, where 57.8% is the infrastructure cost, 23.1% is the active losses cost, and the remaining 19.2% is the reactive losses cost. The wind field $(0_3, N_3)$ has 16 wind turbines, and the total cost is EUR 595,478.1, where 58.1% is the infrastructure cost, 23.5% is the active losses cost, and 18.4% is the reactive losses cost. The wind field $(0_4, N_4)$ has 38 wind turbines, and the total cost is EUR 1,304,863.9, with EUR 758,211.4 for the infrastructure cost, EUR 305,002.4 for the active losses cost, and EUR 241,650.1 for the reactive losses cost. The wind field $(0_5, N_5)$ has 26 wind turbines and the total cost is EUR 1,411,230.1, with the infrastructure cost contributing 57.2% (EUR 807,677.4), the active losses 23.8% (EUR 336,226.6), and the reactive losses cost 18.9% (EUR 267,326.1).

The most expensive wind field is $(0_5, N_5)$, although it has 12 fewer turbines than $(0_4, N_4)$. This fact is due to the position of substation $0_4$, which is more central and has more turbines nearby.

In all case studies, the solutions only present cables of type $k \in \{3, 4, 7, 8, 10\}$.

The *Alto Minho* wind farm results where the turbines were first assigned to the closest substation via the clustering algorithm followed by the ILP model, **Layout+**, are the same as the GA results. This fact is because turbines are grouped closest to one substation and very far from the others.

**Table 6.** Solution description for the *Alto Minho* wind farm.

| Wind Field | $k$ | Links | #Links | Cost (EUR) |
|---|---|---|---|---|
| $0_1$ | 3 | (4, 3), (5, 1), (6, 2), (9, 8), (15, 14), (17, 13), (19, 18), (22, 21), (25, 26) | 9 | $f_1(0_1, N_1) = 1{,}255{,}170.4$ |
| $\lvert N_1 \rvert = 26$ | 4 | $(0_1, 6)$, (7, 4), (7, 5), (10, 9), (16, 15), (20, 17), (20, 19), (23, 22), (24, 25) | 9 | $c_I(0_1, N_1) = 715{,}716.4$ |
| | 7 | $(0_1, 16)$, (0, 23), (11, 10), (12, 24) | 4 | $c_p(0_1, N_1) = 296{,}430.9$ |
| | 8 | $(0_1, 11)$, $(0_1, 12)$ | 2 | $c_q(0_1, N_1) = 243{,}023.2$ |
| | 10 | $(0_1, 7)$, $(0_1, 20)$ | 2 | |
| $0_2$ | 3 | $(0_2, 43)$, (28, 27), (32, 31), (38, 37), (39, 35), (45, 44) | 6 | $f_2(0_2, N_2) = 873{,}066.6$ |
| | 4 | (29, 28), (33, 32), (36, 38), (42, 45) | 4 | $c_I(0_2, N_2) = 503{,}792.0$ |
| $\lvert N_2 \rvert = 19$ | 7 | $(0_2, 33)$, $(0_2, 36)$, (30, 29), (41, 42) | 4 | $c_p(0_2, N_2) = 201{,}297.4$ |
| | 8 | (34, 30), (40, 41) | 2 | $c_q(0_2, N_2) = 167{,}977.2$ |
| | 10 | $(0_2, 34)$, $(0_2, 39)$, (39, 40) | 3 | |
| $0_3$ | 3 | (47, 46), (50, 49), (59, 58), (60, 61) | 4 | $f_3(0_3, N_3) = 595{,}478.1$ |
| | 4 | (48, 47), (52, 50), (60, 59) | 3 | |
| $\lvert N_3 \rvert = 16$ | 7 | (51, 48), (54, 52) | 2 | $c_I(0_3, N_3) = 346{,}205.8$ |
| | 8 | $(0_3, 54)$, (53, 51), (57, 60) | 3 | $c_p(0_3, N_3) = 139{,}633.1$ |
| | 10 | $(0_3, 55)$, $(0_3, 56)$, (55, 53), (56, 57) | 4 | $c_q(0_3, N_3) = 109{,}639.2$ |
| $0_4$ | 3 | $(0_4, 72)$, (64, 63), (65, 62), (70, 69), (71, 68), (76, 77), (79, 78), (85, 86), (87, 92), (89, 91), (93, 94) | 11 | $f_4(0_4, N_4) = 1{,}304{,}863.9$ |
| $\lvert N_4 \rvert = 38$ | 4 | $(0_4, 70)$, $(0_4, 71)$, (65, 64), (75, 76), (80, 79), (84, 85), (84, 87), (88, 89), (88, 93), (74, 75), (81, 80), (73, 74), (82, 81), (90, 65) | 14 | $c_I(0_4, N_4) = 758{,}211.4$ |
| | 7 | (74, 75), (81, 80) | 2 | $c_p(0_4, N_4) = 305{,}002.4$ |
| | 8 | (73, 74), (82, 81), (90, 65) | 3 | $c_q(0_4, N_4) = 241{,}650.1$ |
| | 10 | $(0_4, 67)$, $(0_4, 73)$, $(0_4, 82)$, $(0_4, 83)$, $(0_4, 84)$, (66, 90), (67, 66), (83, 88) | 8 | |
| $0_5$ | 3 | $(0_5, 106)$, (97, 95), (97, 96), (99, 98), (102, 101), (103, 116), (110, 112), (118, 119), (120, 117) | 9 | $f_5(0_5, N_5) = 1{,}411{,}230.2$ |
| | 4 | $(0_5, 102)$, (100, 99), (104, 103), (111, 118), (115, 120) | 5 | $c_I(0_5, N_5) = 807{,}677.4$ |
| $\lvert N_5 \rvert = 26$ | 7 | $(0_5, 100)$, $(0_5, 104)$, (105, 97), (110, 111), (114, 115) | 5 | $c_p(0_5, N_5) = 336{,}226.6$ |
| | 8 | $(0_5, 105)$, (113, 114) | 2 | $c_q(0_5, N_5) = 267{,}326.1$ |
| | 10 | $(0_5, 107)$, $(0_5, 109)$, (107, 108), (108, 110), (109, 113) | 5 | |
| | | | | $f = 5{,}439{,}809.2$ |

### 5.5. Genetic Algorithm versus Clustering Algorithm

This section summarizes the optimization results of previous sections, comparing the two approaches, GA and clustering. Table 7 presents the optimal fitness values and total costs of the solutions for the GA and clustering algorithms in lines "GA" and "Clustering", respectively. It can be observed that the GA obtains better results than the clustering algorithm in the first three wind farms and the same outcome in the last wind farm.

For each wind farm, the GA ran five times, and in all executions, the GA found the same solution. Table 7 also illustrates the minimum, maximum, median, mean, and standard deviation of the number of iterations needed by the GA to reach the final solution. It could be observed that the algorithm needs a small number of iterations to obtain the final solution. In addition, the GA approach achieves up to 0.17% of economic savings compared to the clustering approach.

The proposed method optimizes the problems in a single phase, allowing the method to converge to the global optimum. In contrast, sequential optimization does not avoid the convergence to local optima that often occurs. This phenomenon was observed using the clustering technique, when the turbines were grouped to a substation in the first phase and the connections to the substation were optimized in the second phase.

**Table 7.** Optimization results for genetic algorithm (costs and descriptive measures of the number of iterations) and clustering algorithm (costs).

|  | *Alto da Coutada* | **WF-S3** | **WF-S4** | *Alto Minho* |
|---|---|---|---|---|
| GA (costs) | 4,795,930.7 | 2,838,121.1 | 7,159,067.9 | 5,439,809.2 |
| Minimum | 29 | 91 | 135 | 185 |
| Maximum | 56 | 137 | 326 | 466 |
| Median | 39 | 96 | 171 | 265 |
| Mean | 40.7 | 103.2 | 195.8 | 303.6 |
| Standard deviation | 13.5 | 19.1 | 76.7 | 124.4 |
| Clustering (costs) | 4,800,839.0 | 2,839,945.3 | 7,170,952.2 | 5,439,809.2 |

## 6. Conclusions

In the present study, a GA and an integer programming model were applied to optimize wind farms' electrical cable connections considering several substations. Simultaneously, the algorithm selects the wind turbine group to assign to each substation and finds the optimal layout by solving an ILP model.

The objective criterion is the total cost, including the infrastructure cost, encompassing the digging cost and cable cost, and the cost of energy losses during the expected wind farm lifetime, 20 years, for all the substations. A rooted forest is obtained, a union of disjoint rooted spanning trees with a substation in its root.

For each substation and its wind turbines, a minimum spanning tree model where the substation is the root node, with bounds on the number of turbines in each branch line due to cable capacities, is solved. This model is based on the model presented in [4].

The results show that the use of the GA coupled with optimization models plays an essential role in the planning and design of wind farms with several substations, in particular in the configuration of the distribution network and the assignment of turbines to substations, resulting in significant gains over the medium and long term.

In future work, other meta-heuristics will be considered and compared with the GA's performance.

**Author Contributions:** Conceptualization, E.J.S.P., A.C. and J.B.; methodology, E.J.S.P. and A.C.; software, E.J.S.P. and A.C.; validation, E.J.S.P., A.C. and J.B.; formal analysis, E.J.S.P., A.C. and J.B.; investigation, E.J.S.P., A.C. and J.B.; resources, E.J.S.P., A.C. and J.B.; writing—original draft preparation, E.J.S.P., A.C. and J.B.; writing—review and editing, E.J.S.P., A.C. and J.B. All authors have read and agreed to the published version of the manuscript.

**Funding:** This research received no external funding.

**Data Availability Statement:** Although the main data presented in this study are in the Appendix A, further information is available on request from the authors.

**Conflicts of Interest:** The authors declare no conflict of interest.

## Abbreviations

The following abbreviations are used in this manuscript:

| | |
|---|---|
| CMST | Capacitated Minimum Spanning Tree |
| GA | genetic algorithm |
| ILP | integer linear programming |
| MILP | mixed-integer linear programming |
| WF | wind farm |

## Appendix A. Wind Farm Coordinates

This section presents the coordinates of wind farms WF-S3, WF-S4, and *Alto Minho* in Tables A1–A3. For wind farms WF-S3 and WF-S4, they are expressed in Cartesian coordinates, and the *Alto Minho* wind farm is expressed in WGS84. Substations and wind

turbines are labeled in the column No. The columns Latitude and Longitude show the corresponding coordinates.

**Table A1.** WF-S3 wind farm coordinates (Cartesian).

| No. | Latitude | Longitude | No. | Latitude | Longitude | No. | Latitude | Longitude |
|-----|----------|-----------|-----|----------|-----------|-----|----------|-----------|
| $0_1$ | 1000.0 | 1000.0 | 24 | 187.5 | 2225.0 | 50 | 3800.0 | 1000.0 |
| $0_2$ | 1687.5 | 3187.5 | 25 | 1375.0 | 2225.0 | 51 | 3000.0 | 1812.0 |
| $0_3$ | 2875.0 | 1625.0 | 26 | 1800.0 | 2225.0 | 52 | 3375.0 | 1812.0 |
| 1 | 187.5 | 187.5 | 27 | 1800.0 | 2600.0 | 53 | 3375.0 | 2225.0 |
| 2 | 187.5 | 600.0 | 28 | 1375.0 | 3000.0 | 54 | 3375.0 | 2600.0 |
| 3 | 600.0 | 600.0 | 29 | 1000.0 | 3000.0 | 55 | 3800.0 | 2600.0 |
| 4 | 600.0 | 187.5 | 30 | 600.0 | 3000.0 | 56 | 3800.0 | 187.5 |
| 5 | 1000.0 | 187.5 | 31 | 187.5 | 3000.0 | 57 | 3800.0 | 1410.0 |
| 6 | 1000.0 | 600.0 | 32 | 187.5 | 3410.0 | 58 | 3800.0 | 3000.0 |
| 7 | 1375.0 | 1000.0 | 33 | 1375.0 | 3410.0 | 59 | 3375.0 | 3000.0 |
| 8 | 1375.0 | 600.0 | 34 | 1000.0 | 3410.0 | 60 | 3000.0 | 2225.0 |
| 9 | 1800.0 | 600.0 | 35 | 1375.0 | 3812.0 | 61 | 3000.0 | 2600.0 |
| 10 | 1800.0 | 187.5 | 36 | 1000.0 | 3812.0 | 62 | 3000.0 | 1410.0 |
| 11 | 2600.0 | 187.5 | 37 | 600.0 | 3812.0 | 63 | 3000.0 | 1000.0 |
| 12 | 2187.5 | 600.0 | 38 | 1800.0 | 3410.0 | 64 | 3000.0 | 600.0 |
| 13 | 2187.5 | 1000.0 | 39 | 1800.0 | 3812.0 | 65 | 3375.0 | 600.0 |
| 14 | 1800.0 | 1000.0 | 40 | 2187.5 | 3812.0 | 66 | 3375.0 | 187.5 |
| 15 | 1800.0 | 1410.0 | 41 | 2187.5 | 3410.0 | 67 | 2600.0 | 2225.0 |
| 16 | 1800.0 | 1812.0 | 42 | 2187.5 | 3000.0 | 68 | 2187.5 | 2225.0 |
| 17 | 1375.0 | 1812.0 | 43 | 2187.5 | 2600.0 | 69 | 2187.5 | 1812.0 |
| 18 | 1000.0 | 2225.0 | 44 | 2600.0 | 3000.0 | 70 | 2600.0 | 1410.0 |
| 19 | 600.0 | 1410.0 | 45 | 3000.0 | 3000.0 | 71 | 2187.5 | 1410.0 |
| 20 | 187.5 | 1000.0 | 46 | 3000.0 | 3410.0 | 72 | 2600.0 | 1000.0 |
| 21 | 600.0 | 1812.0 | 47 | 3000.0 | 3812.0 | 73 | 2600.0 | 600.0 |
| 22 | 187.5 | 1410.0 | 48 | 3375.0 | 3410.0 | 74 | 2600.0 | 1812.0 |
| 23 | 187.5 | 1812.0 | 49 | 3800.0 | 3812.0 | | | |

**Table A2.** *WF-S4* wind farm coordinates (Cartesian).

| No. | Latitude | Longitude | No. | Latitude | Longitude | No. | Latitude | Longitude |
|-----|----------|-----------|-----|----------|-----------|-----|----------|-----------|
| $0_1$ | 5985 | 0 | 25 | 10,080 | 1260 | 53 | 8820 | 3780 |
| $0_2$ | 5985 | 5670 | 26 | 0 | 1890 | 54 | 9450 | 3780 |
| $0_3$ | 0 | 2835 | 27 | 630 | 1890 | 55 | 630 | 4410 |
| $0_4$ | 11,970 | 2835 | 28 | 1260 | 1890 | 56 | 2520 | 4410 |
| 1 | 0 | 05 | 29 | 1890 | 1890 | 57 | 3150 | 4410 |
| 2 | 1890 | 0 0 | 30 | 2520 | 1890 | 58 | 3780 | 4410 |
| 3 | 3150 | 0 0 | 31 | 3150 | 1890 | 59 | 4410 | 4410 |
| 4 | 4410 | 0 5 | 32 | 3780 | 1890 | 60 | 5040 | 4410 |
| 5 | 5670 | 0 5 | 33 | 4410 | 1890 | 61 | 5670 | 4410 |
| 6 | 6930 | 0 0 | 34 | 8190 | 1890 | 62 | 8190 | 4410 |
| 7 | 8190 | 0 | 35 | 8820 | 1890 | 63 | 10,080 | 4410 |
| 8 | 9450 | 0 0 | 36 | 9450 | 1890 | 64 | 11,340 | 4410 |
| 9 | 10,080 | 0 0 | 37 | 10,080 | 1890 | 65 | 0 | 5040 |
| 10 | 11,340 | 0 5 | 38 | 2520 | 2520 | 66 | 630 | 5040 |
| 11 | 1890 | 630 5 | 39 | 3150 | 2520 | 67 | 1260 | 5040 |
| 12 | 2520 | 630 0 | 40 | 5670 | 2520 | 68 | 5040 | 5040 |
| 13 | 3150 | 630 | 41 | 11,340 | 2520 | 69 | 5670 | 5040 |
| 14 | 5040 | 630 | 42 | 1260 | 3150 | 70 | 6930 | 5040 |

**Table A2.** *Cont.*

| No. | Latitude | Longitude | No. | Latitude | Longitude | No. | Latitude | Longitude |
|---|---|---|---|---|---|---|---|---|
| 15 | 6930 | 630 | 43 | 1890 | 3150 | 71 | 7560 | 5040 |
| 16 | 8190 | 630 | 44 | 5040 | 3150 | 72 | 8820 | 5040 |
| 17 | 9450 | 630 | 45 | 7560 | 3150 | 73 | 9450 | 5040 |
| 18 | 10,080 | 630 | 46 | 8190 | 3150 | 74 | 10,710 | 5040 |
| 19 | 10,710 | 630 | 47 | 10,080 | 3150 | 75 | 0 | 5670 |
| 20 | 11,970 | 630 | 48 | 10,710 | 3150 | 76 | 1260 | 5670 |
| 21 | 0 | 1260 | 49 | 1260 | 3780 | 77 | 4410 | 5670 |
| 22 | 3150 | 1260 | 50 | 3780 | 3780 | 78 | 10,080 | 5670 |
| 23 | 6930 | 1260 | 51 | 5040 | 3780 | 79 | 11,340 | 5670 |
| 24 | 7560 | 1260 | 52 | 6300 | 3780 | | | |

**Table A3.** *Alto Minho* wind farm coordinates (WGS84).

| No. | Latitude | Longitude | No. | Latitude | Longitude | No. | Latitude | Longitude |
|---|---|---|---|---|---|---|---|---|
| $0_1$ | 42.06812000 | −8.20812000 | 38 | 41.9795820 | −8.55953200 | 80 | 42.0275590 | −8.24460800 |
| $0_2$ | 41.98118500 | −8.55398750 | 39 | 41.9781410 | −8.55709100 | 81 | 42.0252110 | −8.24368200 |
| $0_3$ | 42.00911250 | −8.29478600 | 40 | 41.9693360 | −8.56515400 | 82 | 42.0220860 | −8.24504800 |
| $0_4$ | 42.01750000 | −8.24810000 | 41 | 41.9664830 | −8.56558400 | 83 | 42.0194900 | −8.24478800 |
| $0_5$ | 42.00574800 | −8.42527700 | 42 | 41.9628160 | −8.56775600 | 84 | 42.0176670 | −8.24278100 |
| 1 | 42.06234500 | −8.22093000 | 43 | 41.9786220 | −8.55331000 | 85 | 42.0162820 | −8.24046600 |
| 2 | 42.06061100 | −8.21760300 | 44 | 41.9598050 | −8.57412400 | 86 | 42.0143180 | −8.23761500 |
| 3 | 42.06966000 | −8.21988600 | 45 | 41.9621250 | −8.57310700 | 87 | 42.0189080 | −8.23708900 |
| 4 | 42.06727800 | −8.21781200 | 46 | 41.9999080 | −8.31788900 | 88 | 42.0225180 | −8.23411100 |
| 5 | 42.06482800 | −8.21646400 | 47 | 42.0009200 | −8.31379700 | 89 | 42.0251990 | −8.23237600 |
| 6 | 42.06087800 | −8.21449400 | 48 | 42.0005580 | −8.31057700 | 90 | 42.0089230 | −8.25538300 |
| 7 | 42.06702400 | −8.21233900 | 49 | 42.0036710 | −8.31001200 | 91 | 42.0288660 | −8.23399900 |
| 8 | 42.08295100 | −8.21161000 | 50 | 42.0038960 | −8.30693100 | 92 | 42.0177650 | −8.23455500 |
| 9 | 42.08160400 | −8.20957200 | 51 | 42.0031320 | −8.30424600 | 93 | 42.0204670 | −8.23054900 |
| 10 | 42.07927000 | −8.20931100 | 52 | 42.0066620 | −8.30269800 | 94 | 42.0231690 | −8.22642100 |
| 11 | 42.07651500 | −8.20818400 | 53 | 42.0054850 | −8.30027200 | 95 | 42.0211900 | −8.42995600 |
| 12 | 42.06792900 | −8.20512400 | 54 | 42.0084240 | −8.29911700 | 96 | 42.0221080 | −8.42636000 |
| 13 | 42.09718500 | −8.20167500 | 55 | 42.0081260 | −8.29605300 | 97 | 42.0196790 | −8.42674400 |
| 14 | 42.09437000 | −8.20232700 | 56 | 42.0095310 | −8.29178600 | 98 | 42.0184890 | −8.42476200 |
| 15 | 42.09203400 | −8.20177600 | 57 | 42.0109160 | −8.28841400 | 99 | 42.0171740 | −8.42304700 |
| 16 | 42.08821600 | −8.20194500 | 58 | 42.0183410 | −8.28622900 | 100 | 42.0149590 | −8.42136400 |
| 17 | 42.08619100 | −8.20089400 | 59 | 42.0164640 | −8.28558600 | 101 | 42.0160320 | −8.41794900 |
| 18 | 42.08417600 | −8.20509100 | 60 | 42.0136670 | −8.28629400 | 102 | 42.0140210 | −8.41750800 |
| 19 | 42.08125300 | −8.20571900 | 61 | 42.0139670 | −8.28374200 | 103 | 42.0077070 | −8.43174300 |
| 20 | 42.07943700 | −8.20484700 | 62 | 42.0080580 | −8.25947500 | 104 | 42.0079810 | −8.42899900 |
| 21 | 42.07777800 | −8.20324800 | 63 | 42.0021230 | −8.26095700 | 105 | 42.0080450 | −8.42543600 |
| 22 | 42.07474000 | −8.20288700 | 64 | 42.0044680 | −8.25875600 | 106 | 42.0042880 | −8.42527700 |
| 23 | 42.07185900 | −8.20300800 | 65 | 42.0069610 | −8.25708500 | 107 | 42.0024680 | −8.42325200 |
| 24 | 42.06747100 | −8.20266400 | 66 | 42.0108030 | −8.25362200 | 108 | 42.0007390 | −8.42163800 |
| 25 | 42.06572200 | −8.20121100 | 67 | 42.0125670 | −8.25204400 | 109 | 41.9983780 | −8.41955800 |
| 26 | 42.06410900 | −8.19974500 | 68 | 42.0125660 | −8.24913300 | 110 | 41.9956310 | −8.41785700 |
| 27 | 41.99135700 | −8.57469800 | 69 | 42.0133550 | −8.24308600 | 111 | 41.9935100 | −8.41694600 |
| 28 | 41.99148500 | −8.57044800 | 70 | 42.0148410 | −8.24581000 | 112 | 41.9921770 | −8.41517200 |
| 29 | 41.99038800 | −8.56836700 | 71 | 42.0157250 | −8.24887900 | 113 | 41.9901420 | −8.41345200 |
| 30 | 41.98899400 | −8.56661400 | 72 | 42.0198430 | −8.24780400 | 114 | 41.9883050 | −8.41180300 |
| 31 | 41.98863900 | −8.56326000 | 73 | 42.0235240 | −8.25009200 | 115 | 41.9861220 | −8.40928800 |
| 32 | 41.98721600 | −8.56094000 | 74 | 42.0256560 | −8.25114200 | 116 | 42.0053200 | −8.43361200 |
| 33 | 41.98593000 | −8.55911400 | 75 | 42.0277040 | −8.25301400 | 117 | 41.9809160 | −8.40226200 |
| 34 | 41.98348100 | −8.55917900 | 76 | 42.0290400 | −8.25483500 | 118 | 41.9829660 | −8.41030700 |
| 35 | 41.97579100 | −8.55719200 | 77 | 42.0313290 | −8.25651100 | 119 | 41.9819610 | −8.40752800 |
| 36 | 41.98112700 | −8.55592400 | 78 | 42.0314560 | −8.24149400 | 120 | 41.9817090 | −8.40417400 |
| 37 | 41.98044700 | −8.56189100 | 79 | 42.0298050 | −8.24384500 | | | |

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
