# Peer review of "Wind Farm Cable Connection Layout Optimization Using a Genetic Algorithm and Integer Linear Programming"

_computation, doi:10.3390/computation11120241_

Round 1

Reviewer 1 Report

Comments and Suggestions for Authors

The ideas of this paper are good. However, it still has some deficiencies as follows:

1.     The abstract of the research paper provides a clear and concise overview of the study, effectively conveying the main objectives, methodology, and key findings. However, there are some areas where the abstract could be enhanced:

(1) Specific Results: While the abstract mentions that the results "enhanced the performance of the proposed approach," it could benefit from specific quantitative results or metrics that demonstrate this improvement. Providing concrete numbers or percentages could make the findings more compelling.

(2) Clarity of Hybrid Approach: The abstract mentions the use of a hybrid approach (GA and MILP), but a brief explanation of why this combination is advantageous would enhance reader understanding.

2.     There are a few areas where the introduction could be further refined:

(1) Transitions: Ensure smoother transitions between different sections within the introduction. For example, clearly signaling when you are transitioning from discussing the global energy context to the specific research problem and approach.

(2) Specific Research Contribution: While the introduction mentions that the research contributes by solving the wind farm layout optimization challenge, it could benefit from a more explicit statement of how this research differs from previous studies.

(3) Related work: The Introduction and related work need to be enhanced, about the algorithm and models. These articles may be helpful for improving this paper: two-level principal-agent model for schedule risk control of IT outsourcing project based on genetic algorithm, simulated annealing genetic algorithm based schedule risk management of IT outsourcing project, a hybrid metaheuristic algorithm for the multi-objective location-routing problem in the early post-disaster stage.

3.     The experiment section of the research paper presents a comprehensive account of the methodology and experimental approach used. However, there are several critical shortcomings that need to be addressed to strengthen this section.

(1) Case Study Emphasis: The experiment section appears to place excessive emphasis on a specific case study and delves deeply into the analysis of its results without highlighting particularly interesting or novel findings. While detailed case studies can be valuable, it's crucial to ensure that they are not overly detailed at the expense of broader applicability.

(2) Limited Algorithm Comparison: The section's comparison of the proposed algorithm with only one other algorithm raises concerns about objectivity. To ensure a more balanced evaluation, the research should consider comparing the proposed method with multiple relevant algorithms, including industry-standard or widely accepted approaches. This would provide a more robust assessment of the method's performance and effectiveness.

References like, a bilevel whale optimization algorithm for risk management scheduling of information technology projects considering outsourcing, colony search optimization algorithm using global optimization, 4pl routing problem using hybrid beetle swarm optimization, may be helpful. That is the way to verify the contribution of the paper.

(3) Interpretation of Results: While the section provides extensive analysis of results, it should aim to extract and emphasize significant findings and insights. The interpretation of the results should focus on key takeaways and practical implications, ensuring that readers gain a clear understanding of the research's contributions.

Comments on the Quality of English Language

Minor editing of English language required

Author Response

Reviewer 1
Responses to the reviewer’s comments
Comment 1: Specific Results: While the abstract mentions that the results “enhanced the performance
of the proposed approach,” it could benefit from specific quantitative results or metrics that
demonstrate this improvement. Providing concrete numbers or percentages could make the findings
more compelling.
Response: Dear Reviewer, we appreciate your suggestion, so the improved percentages when compared
to an exact method have been added to the abstract and Section 5.5:
“This methodology is applied to four onshore WFs. The obtained results show that the solution performance
of the proposed approach reaches up to 0.17% of economic savings when compared to the
clustering with ILP approach (an exact approach)”.
“In addition, the GA approach achieves up to 0.17% of economic savings compared to the clustering
approach”.
Comment 2: Clarity of Hybrid Approach: The abstract mentions the use of a hybrid approach (GA
and MILP), but a brief explanation of why this combination is advantageous would enhance reader
understanding.
Response: To address the reviewer’s concerns, we have added the following text at the beginning of
Section 3.
“The combination of the two methods enable to determine the turbines associated with each substation
and the connection of the turbines to each substation. The GA, acting at a higher level, is responsible
for determining the turbines associated with each substation and, at a lower level, the ILP model is
called upon by the fitness function to determine the optimum link between the turbines and their cable
connection types. In this way, the search is performed as a whole.”
Comment 3: Transitions: Ensure smoother transitions between different sections within the introduction.
For example, clearly signaling when you are transitioning from discussing the global energy
context to the specific research problem and approach.
Response: We agree with the reviewer. To improve the transition between the contexts covered in
the introduction, a new paragraph was added:
Optimizing the wind farm distribution grid is crucial for several reasons, and it contributes to the
overall efficiency, reliability, and economic viability of the wind energy system. The best optimization
solutions can maximizing energy production, extracting the maximum amount of energy from the wind
resources. The grid stability and reliability can be enhanced, efficiency improvements resulting from optimization can lead to cost savings. This includes better maintenance planning, reduced downtime,
and increased lifespan of equipment. Additionally, optimized energy production can contribute to a
more cost-effective energy generation process. Many regions have regulations and standards in place to
ensure the stability and reliability of the power grid. Optimizing the wind farm power grid helps meet
these regulatory requirements, avoiding penalties and ensuring compliance.
In summary, optimizing the power grid of a wind farm is essential for maximizing energy production,
ensuring grid stability, reducing costs, meeting regulatory requirements, and advancing the overall
sustainability and reliability of the energy system.
Comment 4: Specific Research Contribution: While the introduction mentions that the research
contributes by solving the wind farm layout optimization challenge, it could benefit from a more explicit
statement of how this research differs from previous studies.
Response: Dear reviewer, thank you for your suggestion. We have revised the paper to emphasize
its main contributions and to highlight the differences from previous works found in the literature.
“This paper’s main contribution is to optimize the layout of wind farms, in one step, considering
multiple substations and cable connections. In contrast to the usual approaches found in the literature,
which address only one singular substation or a reduced number of turbines.”
Comment 5: Related work: The Introduction and related work need to be enhanced, about the
algorithm and models. These articles may be helpful for improving this paper: two-level principalagent
model for schedule risk control of IT outsourcing project based on genetic algorithm, simulated
annealing genetic algorithm based schedule risk management of IT outsourcing project, a hybrid
metaheuristic algorithm for the multi-objective location-routing problem in the early post-disaster
stage.
Response: Dear reviewer, we have added one of the paper suggestions that is in line with the presented
work.
Comment 6: Case Study Emphasis: The experiment section appears to place excessive emphasis on
a specific case study and delves deeply into the analysis of its results without highlighting particularly
interesting or novel findings. While detailed case studies can be valuable, it’s crucial to ensure that
they are not overly detailed at the expense of broader applicability.
Response: Thank for your suggestion. However, we maintain the description of the case studies
to appeal to a wider readership while the innovative part of the paper is now more clearly integrated
into the text. Moreover, the methodology used is described in detail, enabling easy replication and
comparison with new techniques that may be proposed.
Comment 7: Limited Algorithm Comparison: The section’s comparison of the proposed algorithm
with only one other algorithm raises concerns about objectivity. To ensure a more balanced evaluation,
the research should consider comparing the proposed method with multiple relevant algorithms, including
industry-standard or widely accepted approaches. This would provide a more robust assessment
of the method’s performance and effectiveness.
References like, a bilevel whale optimization algorithm for risk management scheduling of information
technology projects considering outsourcing, colony search optimization algorithm using global optimization,
4pl routing problem using hybrid beetle swarm optimization, may be helpful. That is the
way to verify the contribution of the paper.

Response: Dear reviewer, thank you for your concern. Indeed, there are many metaheuristics used
in search and optimization problems. Most of them are inspired by swarms such as the bilevel whale,
the ant colony, the bee swarm, the beetle swarm, and the bat algorithm, among others. However,
algorithms based on swarms, where agents are defined by positions in the search space, do not allow
direct adaptation when associating turbines with substations. Genetic algorithms based on populations
of chromosomes can store symbolic values or discrete variables in a natural way. This representation
allows them to solve the problem intuitively.
Although pertinent, to our knowledge, there is no commercial application that solves the proposed
problem.
Comment 8: Interpretation of Results: While the section provides extensive analysis of results,
it should aim to extract and emphasize significant findings and insights. The interpretation of the
results should focus on key takeaways and practical implications, ensuring that readers gain a clear
understanding of the research’s contributions.
Response: Dear Reviewer, We are adding the following paragraph to address your concerns:
“The proposed method optimizes the problems in a single phase allowing the method to converge to the
global optimum. In contrast, sequential optimization does not avoids the convergence to local optima
that often occurs. This phenomenon was observed, using the clustering technique, when the turbines
were grouped to a substation in the first phase, and the connections to the substation were optimized
in the second phase.”

Reviewer 2 Report

Comments and Suggestions for Authors

This paper deals with the wind farm (WF) optimization layout considering several substations in order to minimize the infrastructure cost and the cost of electrical energy losses during the wind farm lifetime. It is generally well-written; however, the authors are advised to address the following points.

1.     Literature review and state-of-the-art presentation should be enhanced. More papers dealing with the examined issue could be included and analyzed regarding the added value of the present paper, e.g.

F. D. Kanellos and J. Kabouris, "Wind Farms Modeling for Short-Circuit Level Calculations in Large Power Systems," in IEEE Transactions on Power Delivery, vol. 24, no. 3, pp. 1687-1695.

2.     A paragraph analyzing in detail the innovative points of this paper should be included in the revised manuscript.

3.     Analysis regarding the required computation time in relation with the complexity and the size of the Wind Farm should be included.

4.     Why did you choose genetic algorithm and integer linear programming to solve the examined problem?

5.     Can the proposed method be applied to mesh grids or it is suitable only for radial networks?

Comments on the Quality of English Language

Minor editing of English language is required

Author Response

Reviewer 2
Responses to the reviewer’s comments.
Comment 1: Literature review and state-of-the-art presentation should be enhanced. More papers
dealing with the examined issue could be included and analyzed regarding the added value of the
present paper, e.g.
F. D. Kanellos and J. Kabouris, ”Wind Farms Modeling for Short-Circuit Level Calculations in Large
Power Systems,” in IEEE Transactions on Power Delivery, vol. 24, no. 3, pp. 1687-1695.
Response: Dear reviewer, thank you for the suggestion. We have included this reference.
Comment 2: A paragraph analyzing in detail the innovative points of this paper should be included
in the revised manuscript.
Response: Dear Reviewer, at the end of Section 1 you can find the paper’s main contributions. The
following paragraph has been rewritten to highlight the main contributions:
“This paper’s main contribution is to optimize the layout of wind farms, in one step, considering multiple
substations and cable connections. In contrast to the usual approaches found in the literature,
which address only one singular substation or a limited turbine numbers. In the optimization process,
a genetic algorithm is used to determine the topology design and an integer linear programming
model determines the optimal cable connection. The overall objective function minimizes energy losses
and cable installation costs. The case studies presented consider up to five substations and 120 wind
turbines, but the methodology could be extended to higher dimensions.”
In addition, we have added a paragraph before the Conclusions section to highlight the benefit of innovating
the method:
“The proposed method optimizes the problems in a single phase allowing the method to converge to the
global optimum. In contrast, sequential optimization does not avoids the convergence to local optima
that often occurs. This phenomenon was observed, using the clustering technique, when the turbines
were grouped to a substation in the first phase, and the connections to the substation were optimized
in the second phase.”
Comment 3: Analysis regarding the required computation time in relation with the complexity and
the size of the Wind Farm should be included.
Response: Computational time has not been the priority of the wind farm project compared to the
building time project. In addition, depends on the resources used. In any case, the running time of the
ILP is insignificant in the order of seconds, and the average number of generations/iterations for the
GA to reach the result is between 39 and 265, depending on the problem. Instead of time complexity,
we have studied the cycle complexity of the algorithm. Table 7 illustrates these results.

“For each wind farm, the GA ran five times, and in all executions, the GA finds the same solution.
Table 7 also illustrates the minimum, maximum, median, mean, and standard deviation of the number
of iterations needed by the GA to reach the final solution. It could be observed that the algorithm needs
a small number of iteration to obtain the final solution.”
Comment 4: Why did you choose genetic algorithm and integer linear programming to solve the
examined problem?
Response: Since ILP finds the exact solution to the problem, there is no need to include the connections
between turbines and substations in the GA or other metaheuristics. To clarify the reviewer’s
concerns, we have added the following text at the beginning of section 3.
“The combination of the two methods enable to determine the turbines associated with each substation
and the connection of the turbines to each substation. The GA, acting at a higher level, is responsible
for determining the turbines associated with each substation and, at a lower level, the ILP model is
called upon by the fitness function to determine the optimum link between the turbines and their cable
connection types. In this way, the search is performed as a whole.”
Comment 5: Can the proposed method be applied to mesh grids or it is suitable only for radial
networks?
Response: The meshed grids are only justified for offshore wind farms, and even then they are rarely
used because their design and implementation is much more expensive due to the need to oversize the
entire grid. The ILP model proposed in this paper is only applicable to wind farms with radial grids.
2

Round 2

Reviewer 1 Report

Comments and Suggestions for Authors

Please check the grammar of the manuscript.

Comments on the Quality of English Language

Minor editing of English language required